# End-to-End Super-Resolution for Remote-Sensing Images Using an Improved Multi-Scale Residual Network

Hai Huan [1,*], Pengcheng Li [2], Nan Zou [2], Chao Wang [2,3], Yaqin Xie [2], Yong Xie [4] and Dongdong Xu [1]

1   School of Artificial Intelligence, Nanjing University of Information Science & Technology, Nanjing 210044, China; xdd@nuist.edu.cn
2   School of Electronic & Information Engineering, Nanjing University of Information Science & Technology, Nanjing 210044, China; 20181219042@nuist.edu.cn (P.L.); 20201249374@nuist.edu.cn (N.Z.); Chaowang@nuist.edu.cn (C.W.); xyq@nuist.edu.cn (Y.X.)
3   Jiangsu Key Laboratory of Meteorological Observation and Information Processing, Nanjing University of Information Science and Technology, Nanjing 210044, China
4   School of Geographical Sciences, Nanjing University of Information Science & Technology, Nanjing 210044, China; xieyong@nuist.edu.cn
*   Correspondence: haihuan@nuist.edu.cn

**Abstract:** Remote-sensing images constitute an important means of obtaining geographic information. Image super-resolution reconstruction techniques are effective methods of improving the spatial resolution of remote-sensing images. Super-resolution reconstruction networks mainly improve the model performance by increasing the network depth. However, blindly increasing the network depth can easily lead to gradient disappearance or gradient explosion, increasing the difficulty of training. This report proposes a new pyramidal multi-scale residual network (PMSRN) that uses hierarchical residual-like connections and dilation convolution to form a multi-scale dilation residual block (MSDRB). The MSDRB enhances the ability to detect context information and fuses hierarchical features through the hierarchical feature fusion structure. Finally, a complementary block of global and local features is added to the reconstruction structure to alleviate the problem that useful original information is ignored. The experimental results showed that, compared with a basic multi-scale residual network, the PMSRN increased the peak signal-to-noise ratio by up to 0.44 dB and the structural similarity to 0.9776.

**Keywords:** remote sensing; super-resolution reconstruction; pyramidal multi-scale residual network; multi-scale dilation residual block; hierarchical feature fusion structure; complementary block

## 1. Introduction

Image resolution indicates the amount of information contained in an image [1]. A high-resolution (HR) image has a higher pixel density, higher definition characteristics, and more detailed texture information than a low-resolution (LR) image. Whereas image resolution is the number of pixels in an image, spatial resolution indicates the minimum size of ground targets whose details can be distinguished and is used in the field of remote sensing. In remote-sensing images, high spatial resolution enables the changes in surface details to be observed more clearly on a smaller spatial size [2]. In actual scenes, some remote-sensing satellites only provide low-spatial-resolution remote-sensing images that do not meet actual usage requirements. Single-image super-resolution (SISR) reconstruction techniques use software methods to improve the spatial resolution of remote-sensing images without changing the imaging system, which makes the use of these images advantageous [3].

The popular SISR reconstruction techniques are mainly based on conventional algorithms and learning-based algorithms. Conventional algorithms are divided into interpolation-based and sparse-based representation methods. Interpolation-based methods, e.g., the bicubic

interpolation algorithm (Bicubic), are simple to implement and have been extensively used; however, linear models have a poor ability to recover high-frequency information [4]. Sparse-based approaches enhance the ability of linear models to recover high-frequency information by using prior knowledge, but these methods are computationally complex and require enormous amounts of computing resources [5]. Applying deep learning to super-resolution reconstruction is an important method based on learning that involves constructing an end-to-end convolutional neural network (CNN) model to learn the mapping relationship between LR and HR images [6]. Unlike conventional algorithms, it can recover high-frequency information without requiring enormous computing resources; consequently, super-resolution (SR) reconstruction based on deep learning has become a research hotspot.

Recently, extensive results have been obtained through the application of deep-learning CNNs to SISR reconstruction techniques [7]. In 2014, Dong et al. [6] proposed the SR convolutional neural network (SRCNN) model, which was the first to use a neural network to learn end-to-end mapping between LR and HR images; however, the input up-sampled images increased the amount of calculations performed in the network model. Thus, in 2016, Dong et al. [8] proposed the fast super-resolution convolutional neural network (FSRCNN), which used a deconvolution layer to serve as a reconstruction structure to reduce the amount of calculations of the network model. However, this caused checkerboard artifacts [9] in the reconstructed images, owing to pixel overlap. In 2016, Shi et al. [9] proposed an efficient sub-pixel convolutional neural network (ESPCN) to reconstruct images by using sub-pixel convolution and solved the checkerboard artifact problem. However, the network depth was less than five layers, causing the SR images reconstructed by this network to possess poor definition.

In 2015, He et al. [10] proposed the residual network (ResNet), which used a residual structure to solve the problem of the inability to perform training when the number of network layers was large. Therefore, adding a ResNet can increase the number of network layers to enhance the feature extraction capabilities of the network [10]. On this basis, in 2016, Kim et al. [11] proposed the very deep super-resolution (VDSR) network which used numerous residual structures in the SR reconstruction network. Further, in the same year, Kim et al. [12] proposed the deeply recursive convolutional network (DRCN) and applied a recursive neural network structure in SR processing for the first time based on VDSR. However, the reconstructed image effects was poor when the up-sampling factor is eight or higher. Therefore, in 2017, Lai et al. [13] proposed the Laplacian super-resolution network (LapSRN), which obtained better reconstruction effects with a high up-sampling factor based on the method of residual prediction implemented in a step-by-step manner and with the design of a new loss function. The reconstruction effect represents the definition of the SR image [13]. In 2017, Lim et al. [14] proposed the enhanced deep super-resolution (EDSR) model, which eliminated the batch normalisation (BN) operation so that the model size could be increased to improve the quality of the outcome. However, the network model contained numerous parameters and was difficult to train. In 2018, Li et al. [15] proposed the multi-scale residual network (MSRN), which combined local and global features to solve the problem of feature disappearance in the transmission process. However, the up-sampling operation lost the feature information of the original image. In 2019, Hui et al. [16] proposed the information multi-distillation network (IMDN) to construct a lightweight multi-distillation block that could reconstruct SR images rapidly. In 2020, Tian et al. [17] proposed the coarse-to-fine super-resolution convolutional neural network (CFSRCNN) with multiple refining modules to increase the model stability. However, the feature extraction capabilities of these two networks were insufficient, and the reconstruction effects need to be improved.

In summary, the current popular methods usually possess the following shortcomings:

1. Difficulty recurring network models: most SR reconstruction models require operators to have superior training methods; meanwhile, some SR reconstruction models

have many network layers, which require sophisticated hardware equipment. These characteristics make these network models difficult to recur.

2. Inadequate feature utilisation: blindly increasing the number of network layers will aggravate image feature forgetting; however, using only a single up-sampling operation to increase the number of pixels in the final reconstruction stage will cause some of the LR image information to be lost.

In view of the above shortcomings, this report presents a novel multi-scale dilation residual block (MSDRB) and new complementary block (CB) for reconstruction and proposes a new pyramidal multi-scale residual network (PMSRN). Firstly, the dilation convolution combination with multiple dilation rates is used to improve the receptive field and reduce the difficulty of training. Simultaneously, to integrate image features of different scales more effectively, hierarchical residual-like connections (namely, Res2Net [18]) are introduced into the MSDRBs to achieve a more granular multi-scale feature representation. On this basis, to solve the problem of forgetting and underutilising network features as much as possible, the output of each MSDRB layer is used as the input of the hierarchical feature fusion structure (HFFS). Finally, the CB module designed during the reconstruction process can fully utilise the useful information in the original LR image.

The contributions of this study are as follows:

1. A new MSDRB is proposed. This module expresses multi-scale features with finer granularity, increases the receptive field of each network layer, and enhances the ability to detect image features adaptively.
2. To fuse the shallow and deep features, a new reconstruction CB is proposed. This module can fully utilise the useful information in the original LR image, prevent network instability, and improve the network robustness and image reconstruction effect.
3. The proposed PMSRN is easier to train than other networks, since its number of parameters is only 43.33% of that of EDSR, and the module is independent and easy to migrate to other networks for learning.

The remainder of this paper is organised as follows. Section 2 introduces the MSDRB and reconstruction part of the CB module and describes the relevant theoretical analysis. Section 3 presents the experimental results and analyses the effectiveness of the algorithm. Section 4 discusses the practical application effects of PMSRN in different scenarios. Finally, Section 5 outlines the conclusions.

## 2. Materials and Methods

The proposed method is suitable for the SR reconstruction of single LR images. To ensure that the proposed method is universally applicable to images from different sensors, a red–green–blue (RGB) colour model is used to convert all the bands in the image. Because these bands have the same spatial resolution, each low-spatial-resolution image can obtain the corresponding three-channel image to be reconstructed and use it as the network model input.

### 2.1. Network Architecture

The objective of SR reconstruction is to reconstruct a high-definition SR image $I^{SR} \in R^{Wr \times Hr \times C}$ from an LR image $I^{LR} \in R^{W \times H \times C}$ by learning the mapping between LR and HR. The number of RGB space channels $C$ is 3, the LR version of the HR image $I^{HR} \in R^{Wr \times Hr \times C}$ is $I^{LR}$, $W$ and $H$, respectively, represent the width and height of the LR, and $r$ represents the up-sampling factor during SR reconstruction. After the network is trained and learned, the weight coefficient set $\hat{\theta}$ obtained as:

$$\hat{\theta} = \operatorname*{argmin}_{\theta} \frac{1}{N} \sum_{i=1}^{N} L^{SR}(F_{\theta}(I_i^{LR}), I_i^{HR}). \tag{1}$$

Here, according to the MSRN [15], $i$ represents the $i$th image in the $N$th training set, and $L^{SR}$ represents the loss function used in an SR reconstruction network. The gradient-

descent method is used to minimise $L^{SR}$ to obtain the mapping function $F_\theta$ of the optimal model. Several researchers have begun to study the loss function $L^{SR}$ to improve network performance through the innovative $L^{SR}$; however, the network performance improvement is not evident [15]. In order to avoid introducing unnecessary training methods and reduce computations, we finally choose the $L_1$ function. Therefore, the loss function $L^{SR}$ can be defined as:

$$L^{SR}(F_\theta(I_i^{LR}), I_i^{HR}) = \left\| F_\theta(I_i^{LR}) - I_i^{HR} \right\|_1. \tag{2}$$

The PMSRN is an improved version of the MSRN. This architecture can reconstruct SISR images into higher resolution images, mainly through feature extraction and image reconstruction. Figure 1 shows the overall architecture.

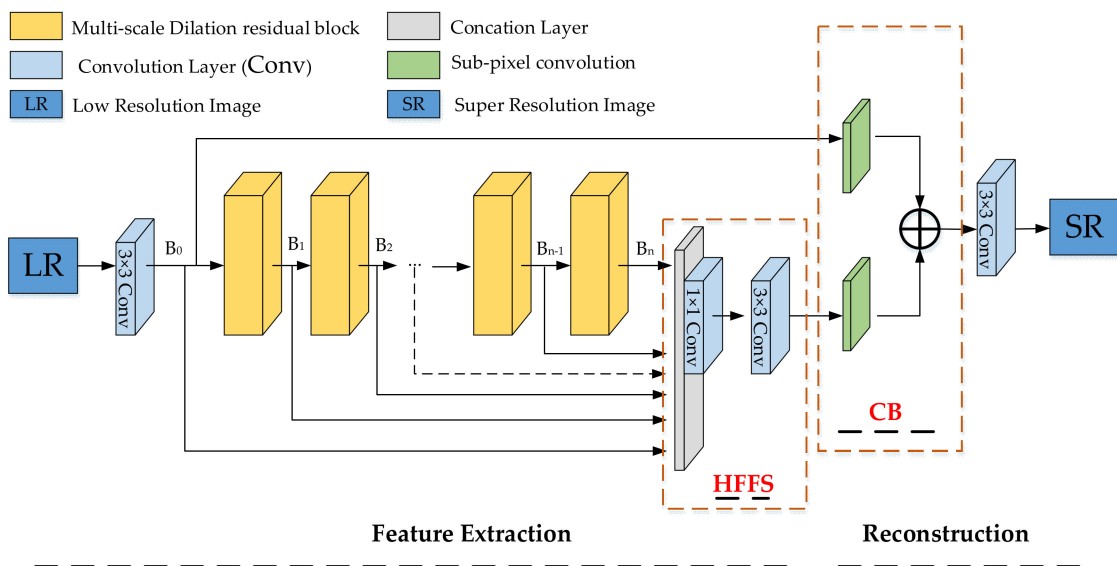

**Figure 1.** Pyramidal multi-scale residual network (PMSRN) model architecture. The network model includes two parts: feature extraction and reconstruction. Feature extraction is performed by the hierarchical feature fusion structure (HFFS) and eight multi-scale dilation residual blocks (MSDRBs), and reconstruction mainly involves a complementary block (CB).

In the training process, firstly, the RGB colour model is used to convert all the bands contained in the public image to obtain the HR image. Secondly, the LR obtained by the HR through the Bicubic down-sampling operation is used as the PMSRN input. Thirdly, the PMSRN uses multiple MSDRBs to learn the feature mapping relationship between the LR and HR. Furthermore, the global and local feature information are subsequently combined through the HFFS. Finally, the CB reconstructs the SR image.

Our approach differs from the original MSRN [15] in two main aspects:

- In the feature extraction, the MSDRB replaces the multi-scale residual module.
- In the reconstruction part, a CB module was added.

### 2.1.1. Multi-Scale Dilation Residual Block (MSDRB)

Firstly, in order to provide the network with stronger multi-scale feature extraction capabilities, an MSDRB (Figure 2) was designed in PMSRN. The MSDRB consists of three parts: multi-scale fusion, multilevel residual learning, and multi-dilation-rate dilated convolution groups.

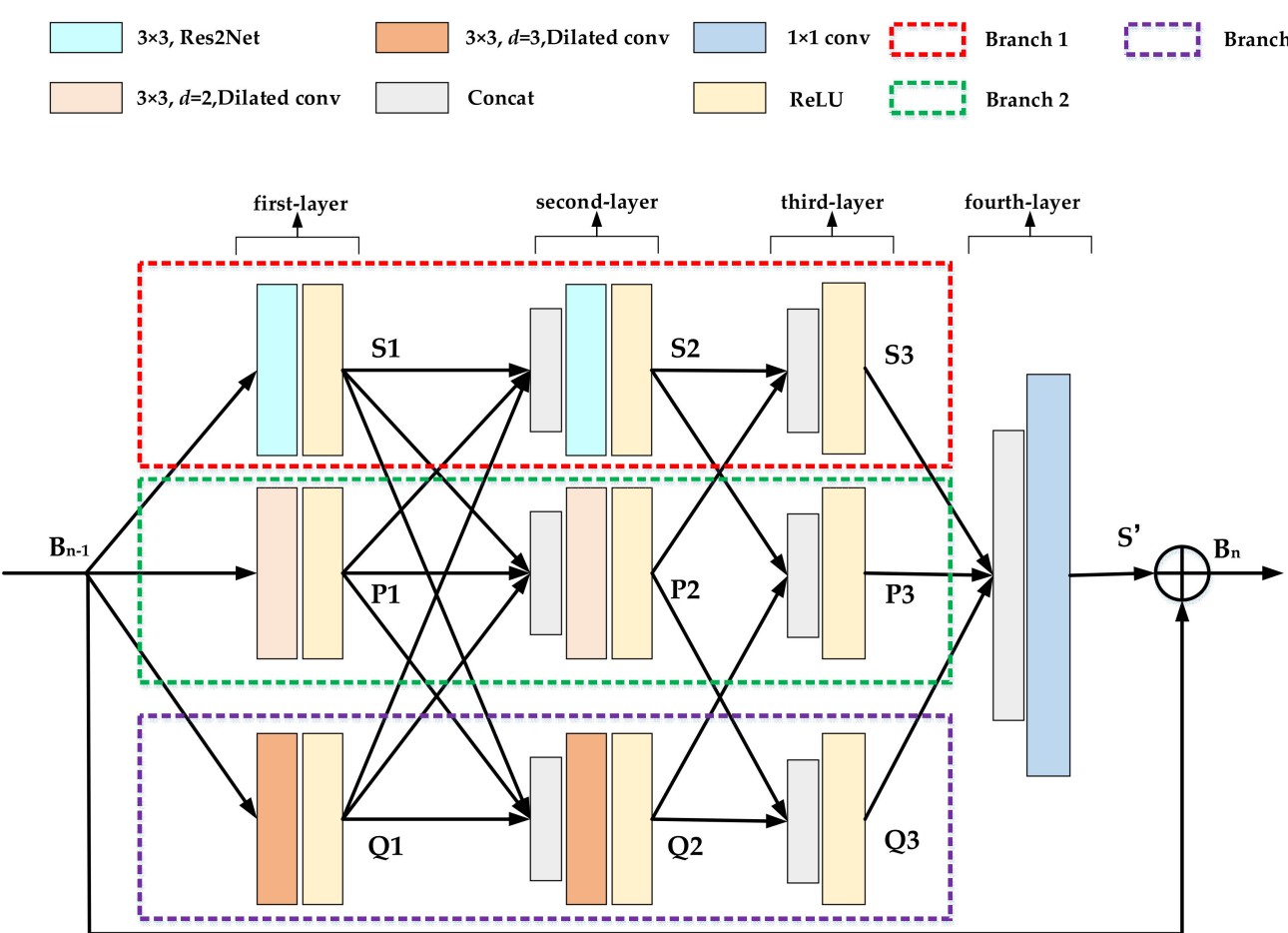

**Figure 2.** MSDRB structure, with four layers divided into three branches. Branch 1 contains the hierarchical residual-like connections (namely, Res2Net) residual block, activation function (i.e., rectified linear unit, ReLU), and concatenation operation. Branches 2 and 3 contain the dilation convolution with dilation rates $d$ of 2 and 3, respectively; ReLU; and concatenation operation. The outputs of branches $S_3$, $P_3$, and $Q_3$ are output as $S'$ after concatenation and $1 \times 1$ convolution. Finally, $S'$ is feature-added with $B_{n-1}$.

Multi-Scale Feature Fusion: the multi-scale nature of the image is similar to that of the human eye observing an object. When the distance from the object is different, the perceived characteristics are different; that is, with the same object in the field of view, the image size and scale are different, so the features are also different [19], multi-scale information is crucial to computer vision algorithms.

In the first-layer network structure, the input $B_{n-1}$ passes through the Res2Net residual block of branch 1, is activated by the activation function rectified linear unit (ReLU, represented by $\sigma$), and outputs $S_1$; input $B_{n-1}$ passes through the dilation convolution of branch 2 (dilation rate $d = 2$), is activated by the ReLU activation function, and outputs $P_1$; input $B_{n-1}$ passes through the dilation convolution of branch 3 (dilation rate $d = 3$), is activated by the ReLU activation function, and outputs $Q_1$. The structure outputs $S_1$, $P_1$, and $Q_1$ can be expressed as follows:

$$S_1 = \sigma(w^1_{3\times3,\text{Res2Net}} * B_{n-1} + b^1), \tag{3}$$

$$P_1 = \sigma(w^1_{3\times3,d=2} * B_{n-1} + b^1), \tag{4}$$

$$Q_1 = \sigma(w^1_{3\times3,d=3} * B_{n-1} + b^1). \tag{5}$$

In the second-layer network structure, the inputs $S_1$, $P_1$, and $Q_1$ output $[S_1, P_1, Q_1]$ through the concatenation operation. $[S_1, P_1, Q_1]$ passes through the Res2Net residual

block of branch 1, is activated by the ReLU activation function, and outputs $S_2$. Then, $[S_1, P_1, Q_1]$ passes through the dilation convolution of branch 2 (expansion rate $d = 2$), is activated by the activation function ReLU, and outputs $P_2$. Further, $[S_1, P_1, Q_1]$ passes through the dilation convolution of branch 3 (expansion rate $d = 3$), is activated by the activation function ReLU, and outputs $Q_2$. The outputs $S_2$, $P_2$, and $Q_2$ of the network structure of the second layer can be expressed as follows:

$$S_2 = \sigma(w^2_{3 \times 3, \text{Res2Net}} * [S_1, P_1, Q_1] + b^2), \tag{6}$$

$$P_2 = \sigma(w^2_{3 \times 3, d=2} * [P_1, S_1, Q_1] + b^2), \tag{7}$$

$$Q_2 = \sigma(w^2_{3 \times 3, d=3} * [Q_1, P_1, S_1] + b^2). \tag{8}$$

In the third-layer network structure, the inputs $S_2$ and $P_2$ of branch 1 output $[S_2, P_2]$ through the concatenation operation, which is activated by the ReLU activation function and outputs $S_3$. The inputs $S_2$ and $Q_2$ of branch 2 output $[S_2, Q_2]$ after the concatenation operation, which is activated by the ReLU activation function and outputs $P_3$. The inputs $Q_2$ and $P_2$ of branch 3 output $[Q_2, P_2]$ after the concatenation operation, which is activated by the ReLU activation function and outputs $Q_3$. The network structure outputs $S_3$, $P_3$ and $Q_3$ of the third layer can be expressed as follows:

$$S_3 = \sigma([S_2, P_2]), \tag{9}$$

$$P_3 = \sigma([S_2, Q_2]), \tag{10}$$

$$Q_3 = \sigma([Q_2, P_2]). \tag{11}$$

In the fourth-layer network structure, the inputs $S_3$, $P_3$, and $Q_3$ are subjected to the concatenation operation; the output is $[S_3, P_3, Q_3]$, which is filtered by a $1 \times 1$ standard convolution kernel and outputs $S'$. The output $S'$ of the fourth layer can be expressed as:

$$S' = \sigma(w^4_{1 \times 1} * [S_3, P_3, Q_3] + b^4). \tag{12}$$

In (3)–(12), $w$ and $b$ represent the weight and bias, respectively; the superscripts represent the numbers of layers; the subscripts $1 \times 1$ and $3 \times 3$ represent the sizes of the convolution kernels; the subscript Res2Net represents the convolution type as hierarchical residual-like connections; and the subscript $d$ represents the dilation convolution with dilation rate $d$.

Let us assume that the number of channels of the MSDRB input $B_{n-1}$ is *n_feats*; then, the numbers of output channels of the internal first, second, and third layers are *n_feats*, $3 \times$ *n_feats*, and $6 \times$ *n_feats*, respectively. In the fourth layer, the number of output channels based on the feature map concatenation is $18 \times$ *n_feats*, and the number of feature map channels is reduced to *n_feats* again with a $1 \times 1$ convolution kernel.

Multilevel Residual Learning: inside a single MSDRB, Res2Net represents multi-scale features with a level that is more granular and increases the receptive field range of each network layer. Specifically, the module replaces a single $3 \times 3$ convolution kernel with a convolution kernel group. Simultaneously, different convolution kernels can be connected in the form of hierarchical residuals (Figure 3b).

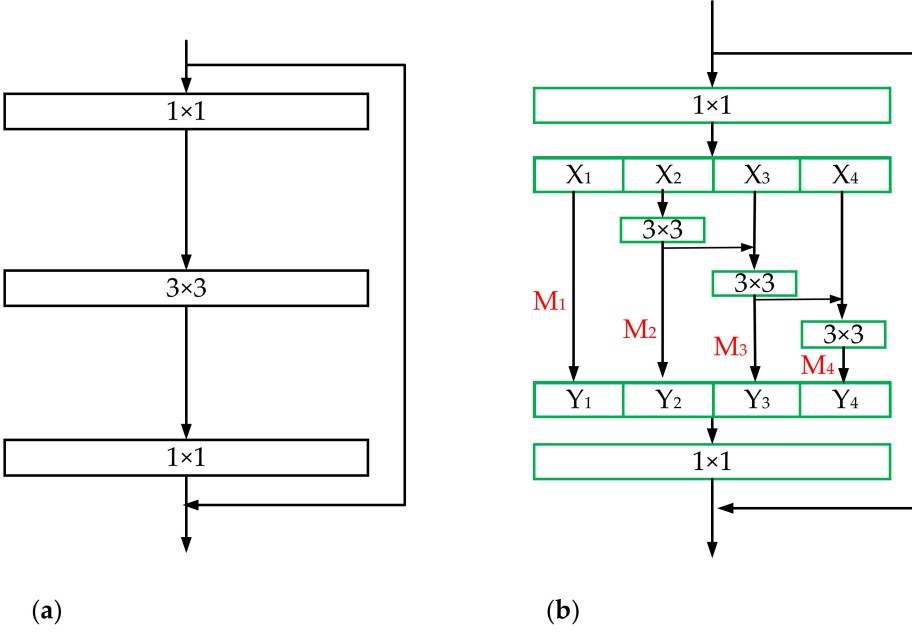

(**a**)                                                                         (**b**)

**Figure 3.** Comparison of ResNet and Res2Net structures. Inside Res2Net, a single $3 \times 3$ convolution kernel is replaced with a convolution kernel group. (**a**) ResNet module and (**b**) Res2Net module.

The internal operation of Res2Net can be defined as:

$$Y_1 = X_1 = M_1, \tag{13}$$

$$Y_2 = X_2 * (3 \times 3) = M_2, \tag{14}$$

$$Y_3 = (M_2 + X_3) * (3 \times 3) = M_3, \tag{15}$$

$$Y_4 = (M_3 + X_4) * (3 \times 3) = M_4. \tag{16}$$

The input image features are filtered by a $1 \times 1$ standard convolution operation and copied into four pieces of feature information, namely, $X_1$, $X_2$, $X_3$, and $X_4$. In Res2Net, the outputs of different receptive fields are obtained. For example, $Y_2$, $Y_3$, and $Y_4$ obtain the receptive fields of the standard convolutions $3 \times 3$, $5 \times 5$, and $7 \times 7$, respectively. Finally, the four outputs are fused, and the number of output channels is reduced to the number of input channels after a $1 \times 1$ convolution operation. This strategy of splitting and fusing enables convolution to process features more efficiently. Note that the MSRN has stronger feature extraction capabilities [15] than the ResNet, dense residual network [20], and inception [21]. To illustrate further the necessity of introducing the Res2Net module, we conducted an experimental comparison with MSRN (discussed in Section 3.1.2).

Outside the MSDRB, add the corresponding elements of $S'$ and $B_{n-1}$ and output $B_n$, which is expressed as:

$$B_n = S' + B_{n-1}. \tag{17}$$

Multi-Dilation Rate Dilated Convolution Group: although this group improves the receptive field and reduces the amount of calculation, the resolution loss is minimised. Dilated convolution is used to set different dilation rates $d$ to obtain different receptive fields. Adding $d - 1$ zeros to the convolution kernel will not increase the amount of calculation. Figure 4 shows dilated convolution kernels at different $d$ values.

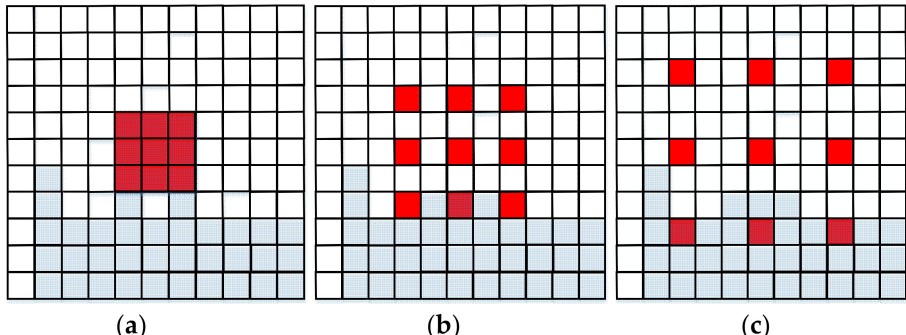

**Figure 4.** Schematic of dilated convolution. Red pixels represent non-zero weight values, white pixels represent zero weight values, and the matrix composed of red pixels is the size of the receptive field: (**a**) $d = 1$, dilation convolution with a receptive field size of $3 \times 3$, the same as that of the standard convolution; (**b**) $d = 2$, dilation convolution with a receptive field size of $5 \times 5$; (**c**) $d = 3$, dilation convolution with a receptive field size of $7 \times 7$.

If the amount of calculation remains constant, different dilation rates $d$ will make the standard convolution $k \times k$ have different receptive fields $R$. The receptive field of dilation convolution [22] can be calculated as follows:

$$R = (d - 1)(k - 1) + k. \tag{18}$$

The calculation based on (18) shows that when $d = 1, 2$, and $3$, the dilation convolution (Figure 4a–c) is equivalent to a $3 \times 3$, $5 \times 5$, and $7 \times 7$ receptive field of the standard convolution, as used in branches 1, 2, and 3 in Figure 2, respectively.

Different branches constructed in this manner have different receptive fields. Combined with the above multi-scale feature fusion and multilevel residual learning, the PMSRN can increase the amount of calculation by a small margin and enhance the ability to detect image feature information (see Section 3.2.3 for a discussion of the corresponding experiments).

### 2.1.2. Complementary Block (CB) in Image Reconstruction Structure

Secondly, in order to fuse the shallow and deep features, a new reconstruction CB is proposed in PMSRN. A CB is constructed in the reconstructed structure of the PMSRN, and its input consists of two parts, namely, original image feature $B_0$ and the HFFS (Figure 5). The HFFS is a global and local feature fusion structure. The inputs $B_0$, $B_1$, ... , and $B_n$ are subjected to the concatenation operation, and the output is filtered through two convolution layers.

$B_0$ and the HFFS respectively perform sub-pixel convolution operations (Figure 6) and rearrange the tensor with dimensions $H \times W \times C \cdot r^2$ as $rH \times rW \times C$. Then, the corresponding elements are added, and the features are reconstructed as SR images after $3 \times 3$ standard convolution filtering. The CB module integrates the global and local features, effectively utilises the original feature information, and prevents information loss. Section 3.1.1 analyses the necessity of the CB module.

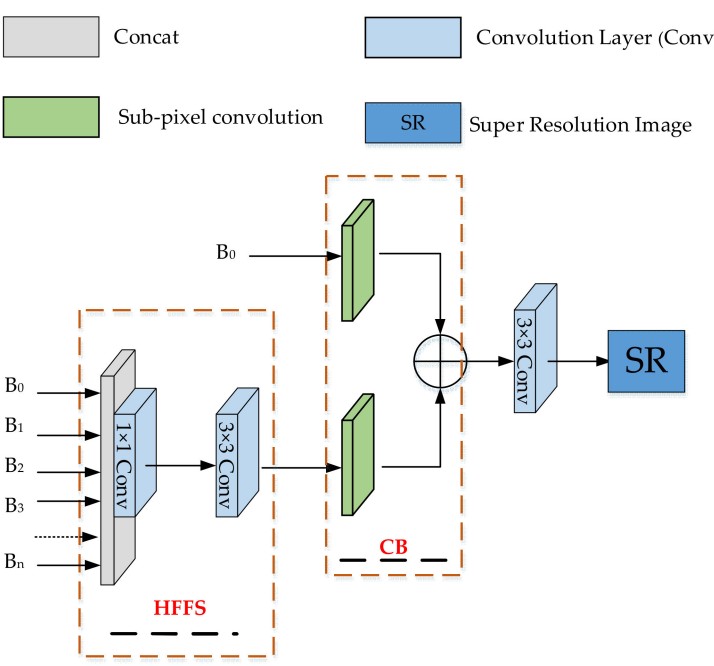

**Figure 5.** Reconstruction structure of the PMSRN, which mainly consists of the CB module. The CB module input includes two parts, namely, the original image feature information $B_0$ and the HFFS. The two parts of the input perform sub-pixel convolution operations, and the corresponding elements are added to reconstruct the SR image. The HFFS concatenates the inputs $B_0$, $B_1$, ... , and $B_n$ and outputs them after two layers of convolutional filtering.

### Sub-Pixel Convolution

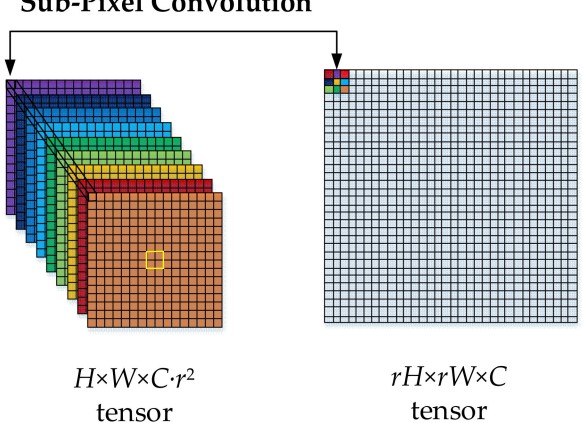

$H \times W \times C \cdot r^2$
tensor

$rH \times rW \times C$
tensor

**Figure 6.** Schematic of the sub-pixel convolution operation. The sub-pixel convolution rearranges the tensor with dimensions of $H \times W \times C \cdot r^2$ to $rH \times rW \times C$.

### 2.2. Datasets

To facilitate experimental comparisons with other SR reconstruction networks, the publicly available, high-quality image dataset, diverse 2K (DIV2K) [23], was selected, which contained 800 training and 100 verification images. In the test phase, the remote-sensing images used in the test stage contained numerous regular roads, buildings, fields, and other features with increased requirements for detailed texture, while the DIV2K training set images showed clear details. By learning the mapping relationship between the LR and HR of the DIV2K training set, the ability of the network model to distinguish detailed textures can be enhanced to improve the spatial resolution of the remote sensing image. Therefore, the DIV2K dataset can be applied to the SR reconstruction of remote-sensing images. The aerial image dataset (AID) is a remote sensing image dataset that includes 30 categories of scene images, each of which has approximately 220–420 pieces (a total of

10,000 pieces), and each image size is 600 × 600. A total of 7614 high-definition images were selected as the new dataset, of which 80% (6768) were used as the training set, 10% (846) as the verification set, and 10% (846) as the AID-test.

The DIV2K dataset is used in the comparative experiments described in Sections 3.1 and 3.2. The AID dataset is used to compare the effect of remote-sensing image reconstruction, as described in Sections 3.3 and 4. Both training sets have 2× (r = 2), 3× (r = 3), 4× (r = 4), and 8× (r = 8) training sets with four different up-sampling factors. To improve the training efficiency, the LR image input after bi-cubic down-sampling was divided into multiple training images with a size of 64 × 64, and sent to the network model for training. Before training, each training block was randomly scaled, rotated, and flipped to increase the training data.

In the test phase of Section 3.2, five public datasets were used: Set5 [24], Set14 [25], BSDS100 [26] (B100), Urban100 [27], and Manga109 [28]. The test sets Set5 and Set14 are low-complexity, single-image, SR reconstruction datasets based on non-negative neighbourhood embedding; BSDS100 and Urban100 comprise 100 images each; and the Manga109 dataset contains 109 high-quality Japanese cartoon images, which can fully verify the performance of the model. Both the PMSRN model and MSRN are trained in RGB space to evaluate their performance. In Section 3.3, the AID-test set will be used as the test set to compare the reconstruction effect of PMSRN on different training sets.

### 2.3. Experimental Environment

The initial learning rate *lr* of the PMSRN network is 0.0001, and the learning rate decreases by 50% every 200 epochs. The optimiser was the Adam optimiser [29]. Eight MS-DRB (n = 8) were used in the model. The number of input channels of each MSDRB was equal to that of the output channels, and the number of output channels of the HFFS module was consistent with that of a single MSDRB. Two NVIDIA GeForce RTX 2080Ti were used to train the PMSRN model on the Pytorch framework. When there is a corresponding HR image, two evaluation standards, peak signal-to-noise ratio (PSNR) [30] and structural similarity (SSIM) [30] are used for evaluation. The higher the PSNR/SSIM value, the better the SR image reconstruction effects.

## 3. Results
### 3.1. Necessity of Introducing CB and Res2Net Modules
### 3.1.1. Benefits of CB

To utilise the global and local feature information fully and enhance the reconstruction effect of SR images, a CB is added to the reconstruction structure of the model. An experiment was performed to analyse the relationship between the peak signal-to-noise ratio (PSNR) of the diverse 2K (DIV2K) verification set and the training batch epoch to confirm the network improvement effected by the CB. Figure 7 compares the PSNRs of the MSRN-CB and FSRCNN, MSRN, and IMDN networks in the epoch range of 0 to 100; curves of different colours are used to indicate different networks. Note that to verify the effectiveness of the CB module, no pre-training parameters (training methods) were used to initialise any of the networks.

The PSNR training curves of the deep network (more than 20 layers) MSRN (blue line), MSRN-CB (green line), and IMDN (purple line) are much higher than that of the shallow network (fewer than 5 layers) FSRCNN (orange line). The MSRN-CB (green line) has the highest PSNR curve and the highest rising speed. With increasing *r*, the PSNR curve of the MSRN-CB becomes increasingly flat, indicating that the CB module can effectively improve the stability and robustness of the network model. To verify further the necessity of introducing the CB module, the Manga109 test set was considered as an example, and Table 1 compares the PSNR/SSIM of the four models.

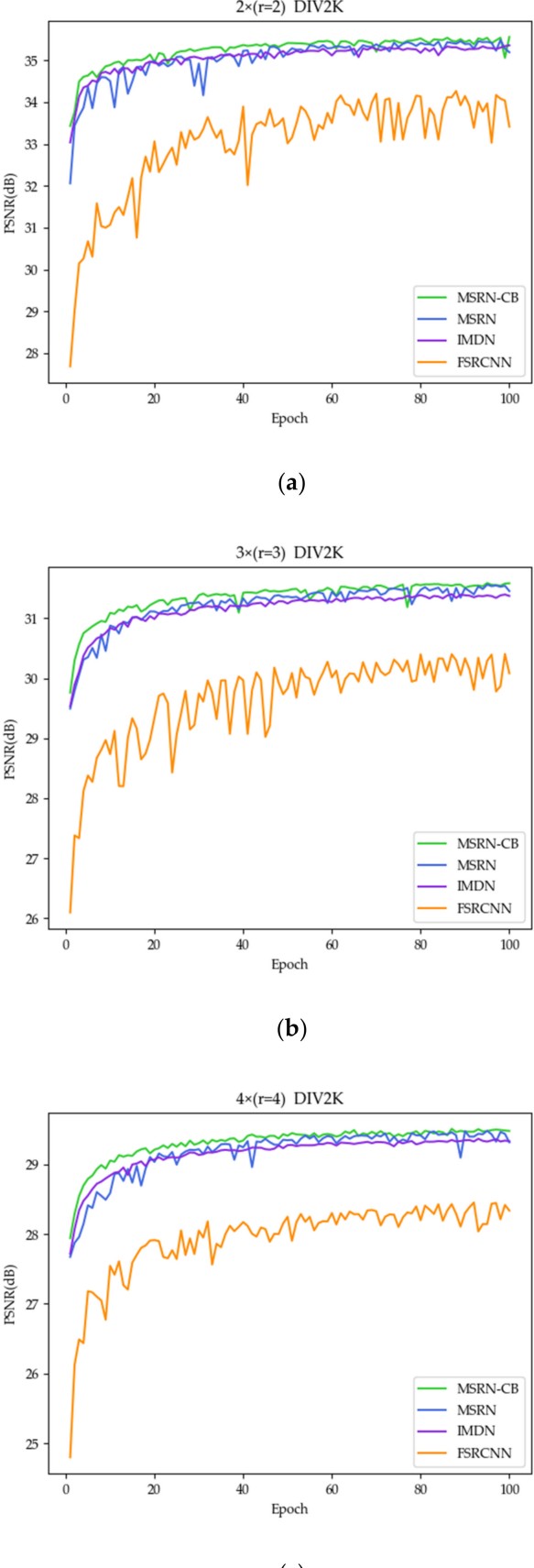

(**a**)

(**b**)

(**c**)

**Figure 7.** *Cont.*

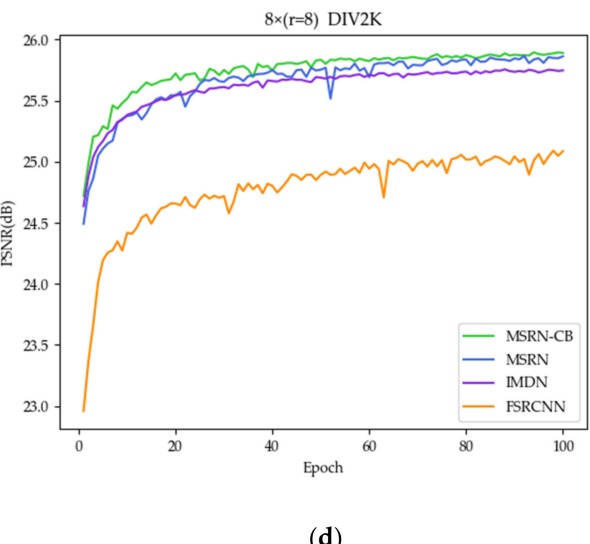

(**d**)

**Figure 7.** Comparison of the peak signal-to-noise ratios (PSNRs) of the multi-scale residual network (MSRN, blue line), MSRN-CB (green line), information multi-distillation network (IMDN, purple line), and fast super-resolution convolutional neural network (FSRCNN, orange line) with training on the diverse 2K (DIV2K) dataset and up-sampling factors *r* values of (**a**) 2, (**b**) 3, (**c**) 4, and (**d**) 8.

**Table 1.** Comparison of MSRN-CB with MSRN, IMDN, and FSRCNN. The numbers in red and blue represent optimal and suboptimal values, respectively.

| Algorithm (Dataset) | Scale | Manga109 PSNR/SSIM |
|:---:|:---:|:---:|
| FSRCNN [8] | 2× | 36.10/0.9695 |
| MSRN [15] | 2× | 38.50/0.9766 |
| MSRN-CB | 2× | 38.50/0.9769 |
| IMDN [16] | 2× | 38.47/0.9766 |
| FSRCNN [8] | 3× | 30.76/0.9188 |
| MSRN [15] | 3× | 33.51/0.9442 |
| MSRN-CB | 3× | 33.63/0.9450 |
| IMDN [16] | 3× | 33.21/0.9420 |
| FSRCNN [8] | 4× | 27.71/0.8633 |
| MSRN [15] | 4× | 30.42/0.9083 |
| MSRN-CB | 4× | 30.49/0.9088 |
| IMDN [16] | 4× | 30.19/0.9042 |
| FSRCNN [8] | 8× | 22.82/0.7048 |
| MSRN [15] | 8× | 24.40/0.7729 |
| MSRN-CB | 8× | 24.43/0.7744 |
| IMDN [16] | 8× | 24.22/0.7656 |

The MSRN-CB has the highest value on the test set Manga109. When *r* = 2, its PSNR is 2.4 dB higher than that of the shallow FSRCNN, 0.03 dB higher than that of the deep IMDN, and the same as that of the deep MSRN, and its SSIM reaches 0.9769. When *r* = 3, its PSNR is 2.87 dB higher than that of the shallow FSRCNN and 0.42 dB and 0.12 dB higher than those of the deep IMDN and MSRN, respectively, and its SSIM reaches 0.9450. When *r* = 4, its PSNR is 2.78 dB higher than that of the shallow FSRCNN and 0.3 dB and 0.08 dB higher than those of the deep IMDN and MSRN, and its SSIM reaches 0.9088. When *r* = 8, its PSNR is 1.61 dB higher than that of the shallow FSRCNN and 0.21 dB and 0.03 dB higher than those of the deep IMDN and MSRN, respectively, and its SSIM reaches 0.7744. It can be seen that as *r* increases, the numerical gap between the MSRN-CB and the other three network models increases, indicating that the MSRN-CB has a better reconstruction effect for larger *r*.

### 3.1.2. Benefits of Res2Net

To increase the receptive field of each network layer and enhance the multi-scale feature extraction capability of the network, Res2Net was introduced as the MSRN-Res2Net network. The experimental method, data, and comparison networks were the same as those in the CB module experiment described in Section 3.1.1, and Figure 8 presents the experimental results.

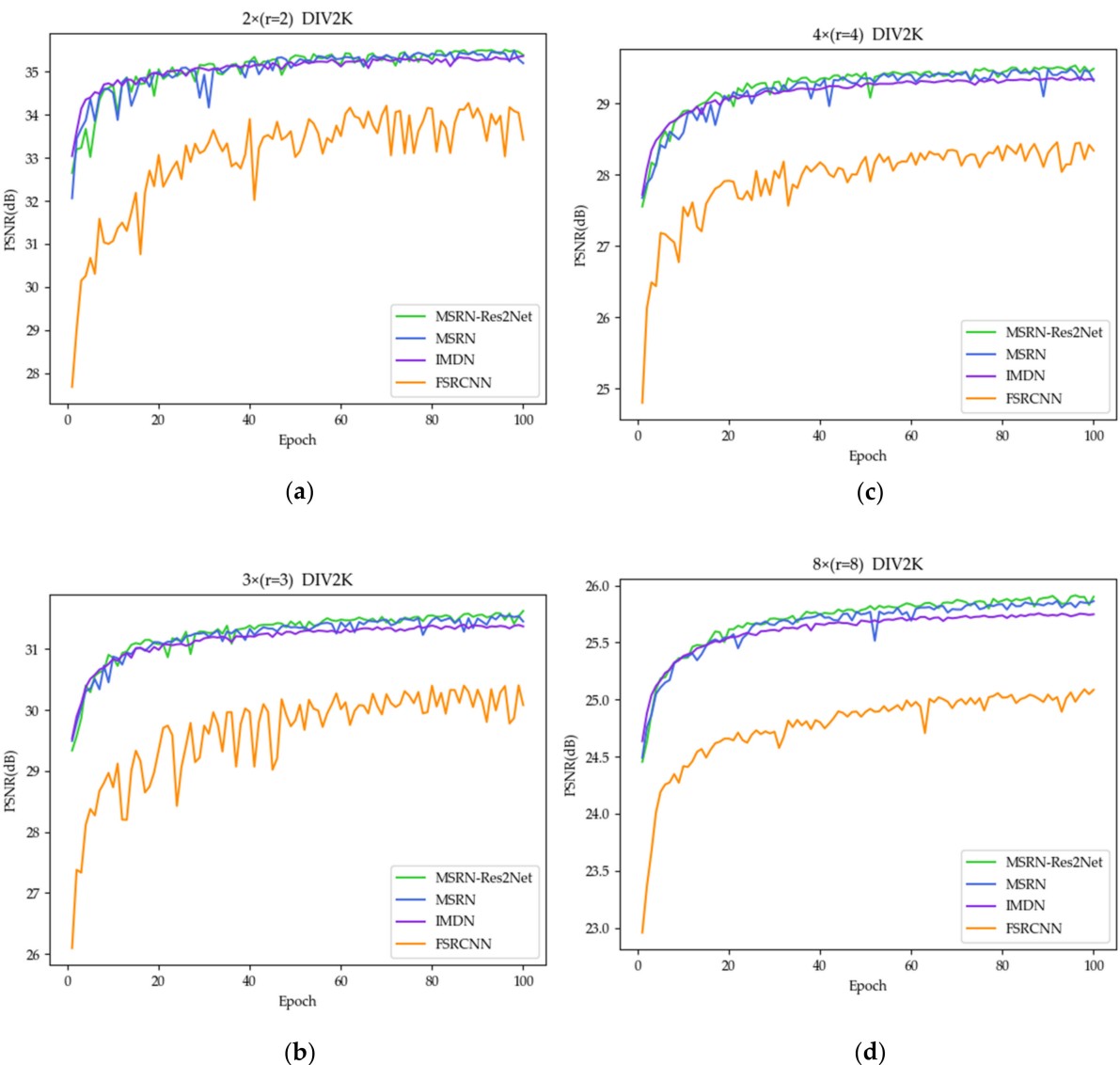

**Figure 8.** Comparison of the PSNRs of the MSRN (blue line), MSRN-Res2Net (green line), IMDN (purple line), and FSRCNN (orange line) with training on the DIV2K dataset and up-sampling factors *r* values of (**a**) 2, (**b**) 3, (**c**) 4, and (**d**) 8.

The PSNR training curves of the deep network MSRN (blue line), MSRN-Res2Net (green line), and IMDN (purple line) are much higher than that of the shallow network FSRCNN (orange line). The MSRN-Res2Net has the highest PSNR curve and highest rising speed. As *r* increases, the distinction between MSRN-Res2Net and the other three curves becomes more obvious. To explain the necessity of adding Res2Net more effectively, Table 2 compares the four models according to the evaluation standard PSNR/SSIM.

**Table 2.** Comparison of MSRN-Res2Net with MSRN, IMDN, and FSRCNN. The numbers in red and blue represent optimal and suboptimal values, respectively.

| Algorithm (Dataset) | Scale | Manga109 PSNR/SSIM |
|---|---|---|
| FSRCNN [8] | 2× | 36.10/0.9695 |
| MSRN [15] | 2× | 38.50/0.9766 |
| MSRN-Res2Net | 2× | 38.68/0.9772 |
| IMDN [16] | 2× | 38.47/0.9766 |
| FSRCNN [8] | 3× | 30.76/0.9188 |
| MSRN [15] | 3× | 33.51/0.9442 |
| MSRN-Res2Net | 3× | 33.60/0.9454 |
| IMDN [16] | 3× | 33.21/0.9420 |
| FSRCNN [8] | 4× | 27.71/0.8633 |
| MSRN [15] | 4× | 30.42/0.9083 |
| MSRN-Res2Net | 4× | 30.65/0.9108 |
| IMDN [16] | 4× | 30.19/0.9042 |
| FSRCNN [8] | 8× | 22.82/0.7048 |
| MSRN [15] | 8× | 24.40/0.7729 |
| MSRN-Res2Net | 8× | 24.55/0.7790 |
| IMDN [16] | 8× | 24.22/0.7656 |

MSRN-Res2Net has the largest value on the test set Manga109. When $r = 2$, its PSNR is 2.58 dB higher than that of the shallow FSRCNN and 0.21 dB and 0.18 dB higher than those of the deep IMDN and MSRN, respectively, and its SSIM reaches 0.9772. When $r = 3$, its PSNR is 2.84 dB higher than that of the shallow FSRCNN and 0.39 dB and 0.09 dB higher than those of the deep IMDN and MSRN, respectively, and its SSIM reaches 0.9454. When $r = 4$, its PSNR is 2.94 dB higher than that of the shallow FSRCNN and 0.46 dB and 0.23 dB higher than those of the deep IMDN and MSRN, respectively, and its SSIM reaches 0.9108. When $r = 8$, its PSNR is 1.73 dB higher than that of the shallow FSRCNN and 0.33 dB and 0.15 dB higher than those of the deep IMDN and MSRN, respectively, and its SSIM reaches 0.7790.

The numerical changes conclusively indicate that as $r$ increases, the numerical gap between the MSRN-Res2Net and the other three network models increases. Thus, the network with Res2Net can also yield better reconstruction results when $r$ is larger.

### 3.2. Comparisons with State-of-the-Art Methods

3.2.1. Comparison of Evaluated Results

The PMSRN was applied to five public datasets and compared with the conventional algorithms and nine state-of-the-art SR methods based on CNNs. These included the Bicubic model, SRCNN [6], FSRCNN [8], ESPCN [9], VDSR [11], DRCN [12], LapSRN [13], EDSR [14], MSRN [15], IMDN [16], and CFSRCNN [17]. Note that the FSRCNN, MSRN, IMDN and PMSRN used the trained network parameters with $r = 2$ to initialise the training networks of other $r$ values and retrain them in the RGB space. For the EDSR [14] and CFSRCNN [17], the data in the corresponding original papers were cited, and for the other methods [6,9,11–13], the comparative data in [15] were cited. Table 3 presents the comparison results. The PMSRN (proposed method) achieved excellent performance for all five public datasets.

Consider the Set5 test set as an example. Compared with other SISR methods, the PMSRN produces superior PSNR and SSIM values. In the SR case with 2× ($r = 2$), the PSNR of the PMSRN is 0.03 dB higher than that obtained with the suboptimal EDSR method and 0.08 dB higher than that of the basic MSRN. In the 3× ($r = 3$) SR case, the PSNR of the PMSRN is 0.01 dB higher than that of the suboptimal EDSR method and 0.21 dB higher than that of the basic MSRN. In the SR case with 4× ($r = 4$), the PSNR of the PMSRN is the same as that of the suboptimal EDSR method and 0.28 dB higher than that of the basic

MSRN. In the SR case with $8\times$ ($r = 8$), the PSNR of the PMSRN is 0.14 dB higher than that of the basic (suboptimal) MSRN.

**Table 3.** Quantitative comparison with state-of-the-art methods. The numbers in red and blue represent optimal and suboptimal values, respectively.

| Algorithm | Scale | Set5 PSNR/SSIM | Set14 PSNR/SSIM | B100 PSNR/SSIM | Urban100 PSNR/SSIM | Manga109 PSNR/SSIM |
|---|---|---|---|---|---|---|
| Bicubic | 2× | 33.69/0.9284 | 30.34/0.8675 | 29.57/0.8434 | 26.88/0.8438 | 30.82/0.9332 |
| SRCNN [6] | 2× | 36.71/0.9536 | 32.32/0.9052 | 31.36/0.8880 | 29.54/0.8962 | 35.74/0.9661 |
| FSRCNN [8] | 2× | 36.89/0.9559 | 32.62/0.9085 | 31.42/0.8895 | 29.73/0.8996 | 36.10/0.9695 |
| ESPCN [9] | 2× | 37.00/0.9559 | 32.75/0.9098 | 31.51/0.8939 | 29.87/0.9065 | 36.21/0.9694 |
| VDSR [11] | 2× | 37.53/0.9583 | 33.05/0.9107 | 31.92/0.8965 | 30.79/0.9157 | 37.22/0.9729 |
| DRCN [12] | 2× | 37.63/0.9584 | 33.06/0.9108 | 31.85/0.8947 | 30.76/0.9147 | 37.63/0.9723 |
| LapSRN [13] | 2× | 37.52/0.9581 | 33.08/0.9109 | 31.80/0.8949 | 30.41/0.9112 | 37.27/0.9855 |
| EDSR [14] | 2× | 38.11/0.9601 | 33.92/0.9195 | 32.32/0.9013 | -/- | -/- |
| MSRN [15] | 2× | 38.06/0.9605 | 33.59/0.9177 | 32.19/0.8999 | 32.10/0.9285 | 38.42/0.9767 |
| IMDN [16] | 2× | 37.89/0.9602 | 33.42/0.9164 | 32.09/0.8985 | 31.84/0.9256 | 38.41/0.9766 |
| CFSRCNN [17] | 2× | 37.79/0.9591 | 33.51/0.9165 | 32.11/0.8988 | 32.07/0.9273 | -/- |
| PMSRN (our) | 2× | 38.14/0.9610 | 33.85/0.9204 | 32.27/0.9007 | 32.51/0.9317 | 38.86/0.9776 |
| Bicubic | 3× | 30.41/0.8655 | 27.64/0.7722 | 27.21/0.7344 | 24.46/0.7411 | 26.96/0.8555 |
| SRCNN [6] | 3× | 32.47/0.9067 | 29.23/0.8201 | 28.31/0.7832 | 26.25/0.8028 | 30.59/0.9107 |
| FSRCNN [8] | 3× | 33.03/0.9141 | 29.46/0.8253 | 28.47/0.7887 | 26.38/0.8065 | 30.87/0.9198 |
| ESPCN [9] | 3× | 33.02/0.9135 | 29.49/0.8271 | 28.50/0.7937 | 26.41/0.8161 | 30.79/0.9181 |
| VDSR [11] | 3× | 33.68/0.9201 | 29.86/0.8312 | 28.83/0.7966 | 27.15/0.8315 | 32.01/0.9310 |
| DRCN [12] | 3× | 33.85/0.9215 | 29.89/0.8317 | 28.81/0.7954 | 27.16/0.8311 | 32.31/0.9328 |
| LapSRN [13] | 3× | 33.82/0.9207 | 29.89/0.8304 | 28.82/0.7950 | 27.07/0.8298 | 32.21/0.9318 |
| EDSR [14] | 3× | 34.65/0.9282 | 30.52/0.8462 | 29.25/0.8093 | -/- | -/- |
| MSRN [15] | 3× | 34.45/0.9276 | 30.40/0.8431 | 29.12/0.8059 | 28.29/0.8549 | 33.62/0.9451 |
| IMDN [16] | 3× | 34.29/0.9266 | 30.23/0.8400 | 29.04/0.8037 | 28.05/0.8498 | 33.32/0.9429 |
| CFSRCNN [17] | 3× | 34.24/0.9256 | 30.27/0.8410 | 29.03/0.8035 | 28.04/0.8496 | -/- |
| PMSRN (our) | 3× | 34.66/0.9291 | 30.48/0.8456 | 29.20/0.8083 | 28.59/0.8616 | 33.92/0.9474 |
| Bicubic | 4× | 28.43/0.8022 | 26.10/0.6936 | 25.97/0.6517 | 23.14/0.6599 | 24.91/0.7826 |
| SRCNN [6] | 4× | 30.50/0.8573 | 27.62/0.7453 | 26.91/0.6994 | 24.53/0.7236 | 27.66/0.8505 |
| FSRCNN [8] | 4× | 30.74/0.8702 | 27.68/0.7580 | 26.97/0.7144 | 24.59/0.7294 | 27.87/0.8650 |
| ESPCN [9] | 4× | 30.66/0.8646 | 27.71/0.7562 | 26.98/0.7124 | 24.60/0.7360 | 27.70/0.8560 |
| VDSR [11] | 4× | 31.36/0.8796 | 28.11/0.7624 | 27.29/0.7167 | 25.18/0.7543 | 28.83/0.8809 |
| DRCN [12] | 4× | 31.56/0.8810 | 28.15/0.7627 | 27.24/0.7150 | 25.15/0.7530 | 28.98/0.8816 |
| LapSRN [13] | 4× | 31.54/0.8811 | 28.19/0.7635 | 27.32/0.7162 | 25.21/0.7564 | 29.09/0.8845 |
| EDSR [14] | 4× | 32.46/0.8968 | 28.80/0.7876 | 27.71/0.7420 | -/- | -/- |
| MSRN [15] | 4× | 32.18/0.8951 | 28.66/0.7835 | 27.61/0.7373 | 26.17/0.7887 | 30.53/0.9093 |
| IMDN [16] | 4× | 32.07/0.8933 | 28.52/0.7800 | 27.52/0.7345 | 25.99/0.7825 | 30.25/0.9052 |
| CFSRCNN [17] | 4× | 32.06/0.8920 | 28.57/0.7800 | 27.53/0.7333 | 26.03/0.7824 | -/- |
| PMSRN (our) | 4× | 32.46/0.8982 | 28.76/0.7863 | 27.69/0.7403 | 26.47/0.7982 | 30.96/0.9146 |
| Bicubic | 8× | 24.40/0.6045 | 23.19/0.5110 | 23.67/0.4808 | 20.74/0.4841 | 21.46/0.6138 |
| SRCNN [6] | 8× | 25.34/0.6471 | 23.86/0.5443 | 24.14/0.5043 | 21.29/0.5133 | 22.46/0.6606 |
| FSRCNN [8] | 8× | 25.82/0.7183 | 24.18/0.6075 | 24.32/0.5729 | 21.56/0.5613 | 22.83/0.7047 |
| ESPCN [9] | 8× | 25.75/0.6738 | 24.21/0.5109 | 24.37/0.5277 | 21.59/0.5420 | 22.83/0.6715 |
| VDSR [11] | 8× | 25.73/0.6743 | 23.20/0.5110 | 24.34/0.5169 | 21.48/0.5289 | 22.73/0.6688 |
| DRCN [12] | 8× | 25.93/0.6743 | 24.25/0.5510 | 24.49/0.5168 | 21.71/0.5289 | 23.20/0.6686 |
| LapSRN [13] | 8× | 26.15/0.7028 | 24.45/0.5792 | 24.54/0.5293 | 21.81/0.5555 | 23.39/0.7068 |
| MSRN [15] | 8× | 26.93/0.7730 | 24.86/0.6388 | 24.78/0.5959 | 22.40/0.6144 | 24.45/0.7746 |
| IMDN [16] | 8× | 26.72/0.7642 | 24.85/0.6363 | 24.74/0.5935 | 22.32/0.6096 | 24.29/0.7680 |
| PMSRN (our) | 8× | 27.07/0.7803 | 24.99/0.6439 | 24.86/0.5995 | 22.60/0.6246 | 24.80/0.7865 |

Next, consider Set14 as an example. In the SR case with $2\times$ ($r = 2$), the SSIM of PMSRN is the best, and the PSNR value of the PMSRN is suboptimal. The PSNR is 0.07 dB lower than that of the optimal EDSR method, and the PSNR of the MSRN is 0.26 dB lower than that of the PMSRN. In the case of SR with $3\times$ ($r = 3$), the SSIM and PSNR of the PMSRN are

both suboptimal. The PSNR is 0.04 dB lower than that of the optimal EDSR method and 0.08 dB higher than that of the basic MSRN. In the case of SR with 4× ($r = 4$), the SSIM and PSNR of the PMSRN are both suboptimal, and the PSNR is 0.04 dB lower than that of the optimal EDSR method and 0.1 dB higher than that of the basic MSRN. In the SR case with 8× ($r = 8$), the PSNR of the PMSRN is 0.13 dB higher than that of the basic (suboptimal) MSRN, and the SSIM of the PMSRN is the best.

Using B100 as an example, in the SR cases with 2× ($r = 2$), 3× ($r = 3$), and 4× ($r = 4$), the PSNRs and SSIMs of the PMSRN are both suboptimal, although the PSNR of the PMSRN is 0.08 dB higher than that of the basic MSRN. In the cases of SR with 2× ($r = 2$) and 3× ($r = 3$), the PSNRs of the PMSRN are 0.05 dB lower than those of the optimal EDSR method. Moreover, in the case of SR with 4× ($r = 4$), the PSNR of the PMSRN is 0.02 dB lower than that of the optimal EDSR method. In the case of SR with 8× ($r = 8$), the PSNR and SSIM of the PMSRN are both optimal, and its PSNR is 0.08 dB higher than that of the basic MSRN (suboptimal).

Taking Urban100 as an example, the PSNR and SSIM of the PMSRN are both optimal compared with those of other SISR methods. In the cases of SR with 2× ($r = 2$), 3× ($r = 3$) and 4× ($r = 4$), and 8× ($r = 8$), the PSNR of the PMSRN is 0.41 dB, 0.3 dB, and 0.2 dB higher than that of the basic (suboptimal) MSRN, respectively.

Considering Manga109 as an example, the PSNR and SSIM of the PMSRN are both optimal compared with those of other SISR methods. In the cases of SR with 2× ($r = 2$), 3× ($r = 3$), 4× ($r = 4$), and 8× ($r = 8$), the PSNR of the PMSRN is 0.44 dB, 0.3 dB, 0.43 dB, and 0.35 dB higher than that of the basic (suboptimal) MSRN, respectively.

Among all $r$ values, compared with the conventional Bicubic technique, the reconstruction network using deep learning yielded the maximum increase in PSNR of 8.04 dB, and the maximum SSIM was 0.9776. Compared with the shallow network SRCNN and FSRCNN, the PMSRN exhibited a significant improvement in SSIM, with an average improvement in PSNR of ~1.3 dB. Compared with the deep network, the PSNR/SSIM of PMSRN were only slightly lower than those of EDSR on test sets Set14 and B100, but training the EDSR model required more memory and space. In contrast, the number of parameters in our model was much smaller than the EDSR model. For more details, please refer to Section 3.2.3.

To show that the PMSRN has an excellent reconstruction effect on images, the visual effect was further analysed quantitatively.

### 3.2.2. Visual Effect Comparison

Figure 9 visually compares the PMSRN with the basic MSRN and conventional Bicubic technique, which are among the methods listed in Table 3. It can be clearly observed that compared with the Bicubic technique, the PMSRN and MSRN yield higher SR image definition.

In the enlarged region, it is apparent that PMSRN produces finer details, such as scarf stripes in 2× ($r = 2$, Figure 9a) and that the reconstructed SR image contains details that closely match the detailed information in the HR image. In the images with larger $r$, the performance of the PMSRN is prominent. For example, in the upper-left red frame of the enlarged areas in the SR images with $r = 3$ and $r = 4$ (Figure 9c,d), the images reconstructed by the PMSRN are similar to the HR images. However, when $r = 4$, the MSRN reconstructs stripes with incorrect detail. In the red frame area with $r = 8$ (Figure 9d), the reconstructed image effects of the PMSRN is substantially higher than that of the Bicubic method. Compared with the MSRN, the PMSRN shows more detailed edge information. The excellent performance of the PMSRN in terms of visual effects is consistent with the quantitative analysis results in Section 3.2.1.

### 3.2.3. Comparison of Network Scales

Figure 10 compares our method with the state-of-the-art models when applied to Set5 (2×, $r = 2$) with respect to the PSNR and the number of parameters.

It can be observed that the number of parameters of PMSRN has only half of the EDSR model, but that the PMSRN performs the best in terms of the PSNR. This finding demonstrates that our model has a more effective structure and achieves a better balance between performance and model size.

### 3.3. Comparison of Reconstruction Effects of Different Training Sets

To study the effects of the DIV2K and AID training sets on the PMSRN model, comparative experiments were conducted on the evaluation results and subjective visual effects. Using 846 remote-sensing images from the AID training set as the test set, the model reconstruction effects of the same algorithm when trained on the DIV2K and AID training sets were compared. Table 4 summarises the evaluation results in terms of the PSNR/SSIM. The names of the datasets are given in parentheses.

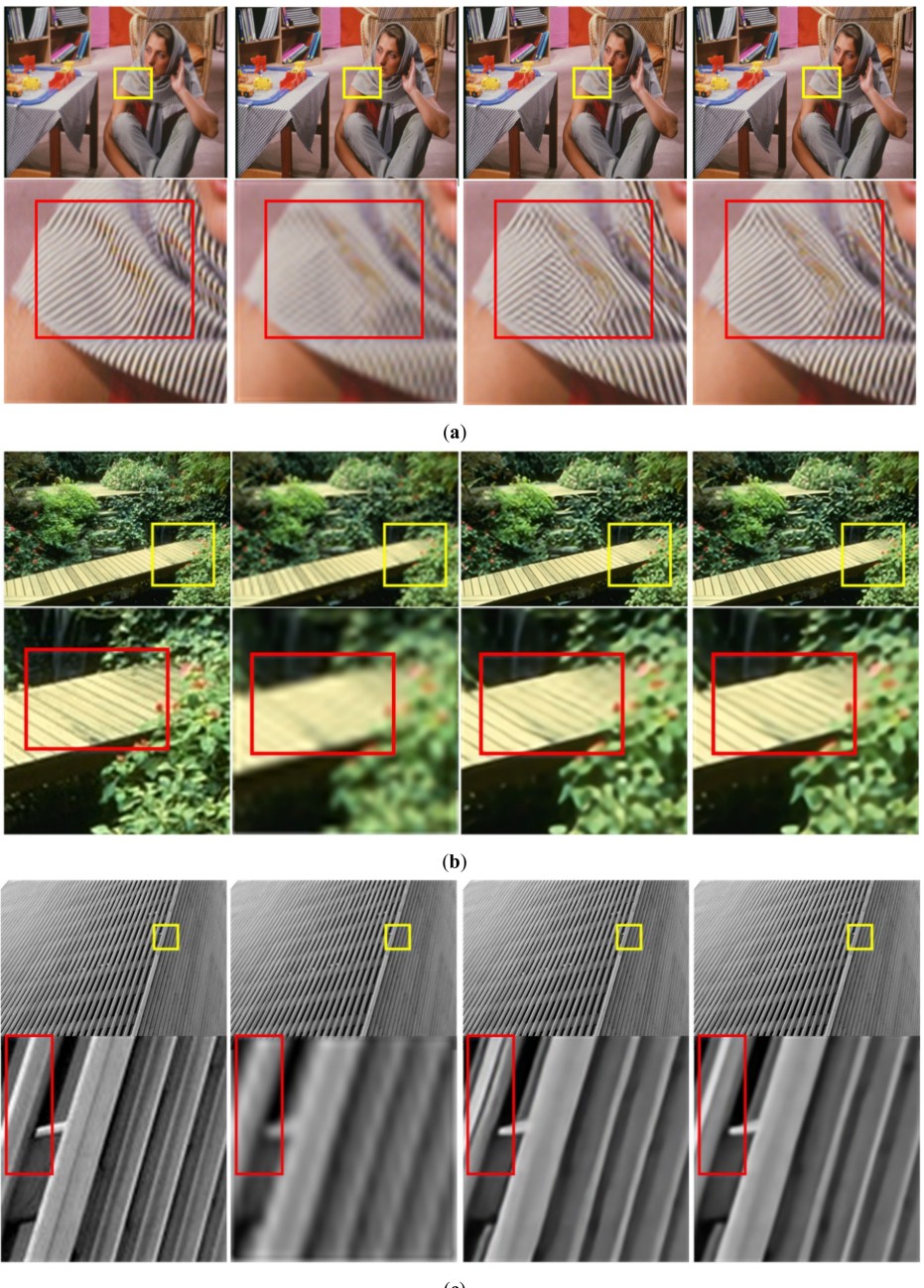

(a)

(b)

(c)

**Figure 9.** *Cont.*

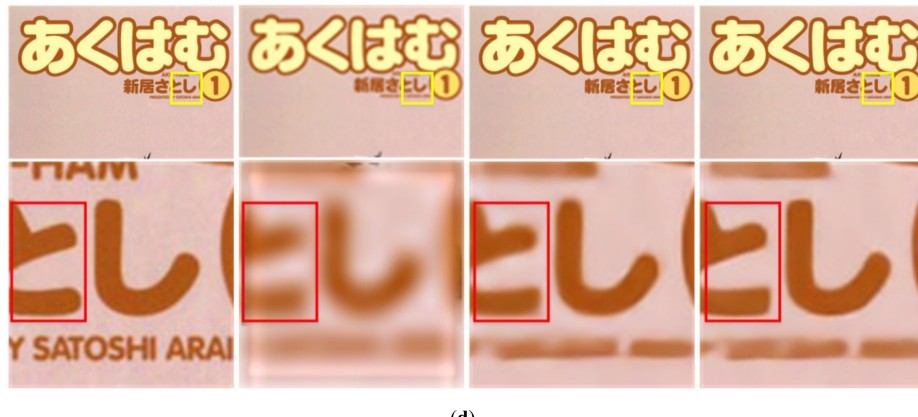

(**d**)

**Figure 9.** Visual comparison of the super-resolution (SR) images obtained using the PMSRN, MSRN, and Bicubic technique using DIV2K as the training set and different *r* values. Each group of images is shown from left to right, including the HR, Bicubic, MSRN, and PMSRN images. Compared with the other methods, the PMSRN produced an SR image with higher definition. The up-sampling factors *r* values are (**a**) 2, (**b**) 3, (**c**) 4, and (**d**) 8.

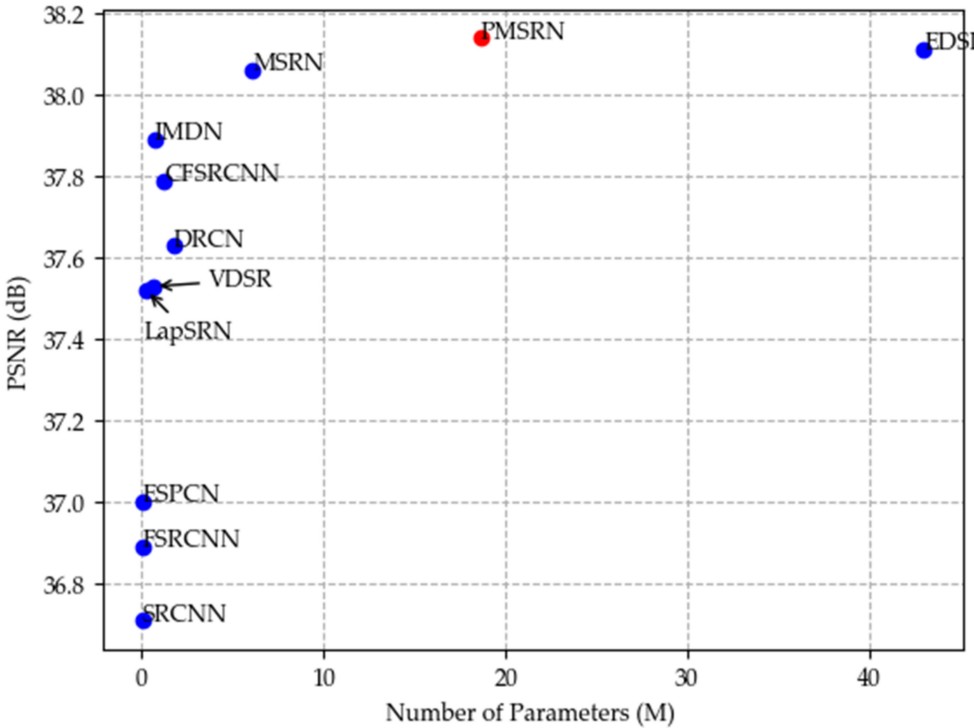

**Figure 10.** Comparison of our method with the state-of-the-art models when applied to Set5 (2×, *r* = 2) with respect to PSNR and the number of parameters ('M' represents the number of parameters in millions).

Although the resolution of each HR image in the DIV2K training set is higher than that in the AID training set, the PSNRs and SSIMs of the PMSRN (DIV2K) are lower than those of the PMSRN (AID) and even lower than those of the MSRN (AID). Therefore, when reconstructing remote-sensing images, the PMSRN (AID) had a better reconstruction effect.

Several groups of images were randomly selected to compare the visual effects of the SR images reconstructed by the PMSRN (DIV2K) and PMSRN (AID) (Figure 11).

**Table 4.** Comparison of SR image evaluation results of different training set reconstruction models. The numbers in red and blue represent optimal and suboptimal values, respectively.

| Algorithm (Dataset) | Scale | AID-Test PSNR/SSIM |
|---|---|---|
| MSRN(DIV2K) | 2× | 35.55/0.9403 |
| MSRN(AID) | 2× | <span style="color:blue">35.91/0.9436</span> |
| PMSRN(DIV2K) | 2× | 35.65/0.9413 |
| PMSRN(AID) | 2× | <span style="color:red">36.00/0.9444</span> |
| MSRN(DIV2K) | 3× | 31.50/0.8634 |
| MSRN(AID) | 3× | <span style="color:blue">31.89/0.8711</span> |
| PMSRN(DIV2K) | 3× | 31.58/0.8656 |
| PMSRN(AID) | 3× | <span style="color:red">32.02/0.8737</span> |
| MSRN(DIV2K) | 4× | 29.30/0.7917 |
| MSRN(AID) | 4× | <span style="color:blue">29.68/0.8028</span> |
| PMSRN(DIV2K) | 4× | 29.39/0.7950 |
| PMSRN(AID) | 4× | <span style="color:red">29.79/0.8068</span> |
| MSRN(DIV2K) | 8× | 25.67/0.6299 |
| MSRN(AID) | 8× | <span style="color:blue">25.87/0.6399</span> |
| PMSRN(DIV2K) | 8× | 25.74/0.6342 |
| PMSRN(AID) | 8× | <span style="color:red">25.93/0.6440</span> |

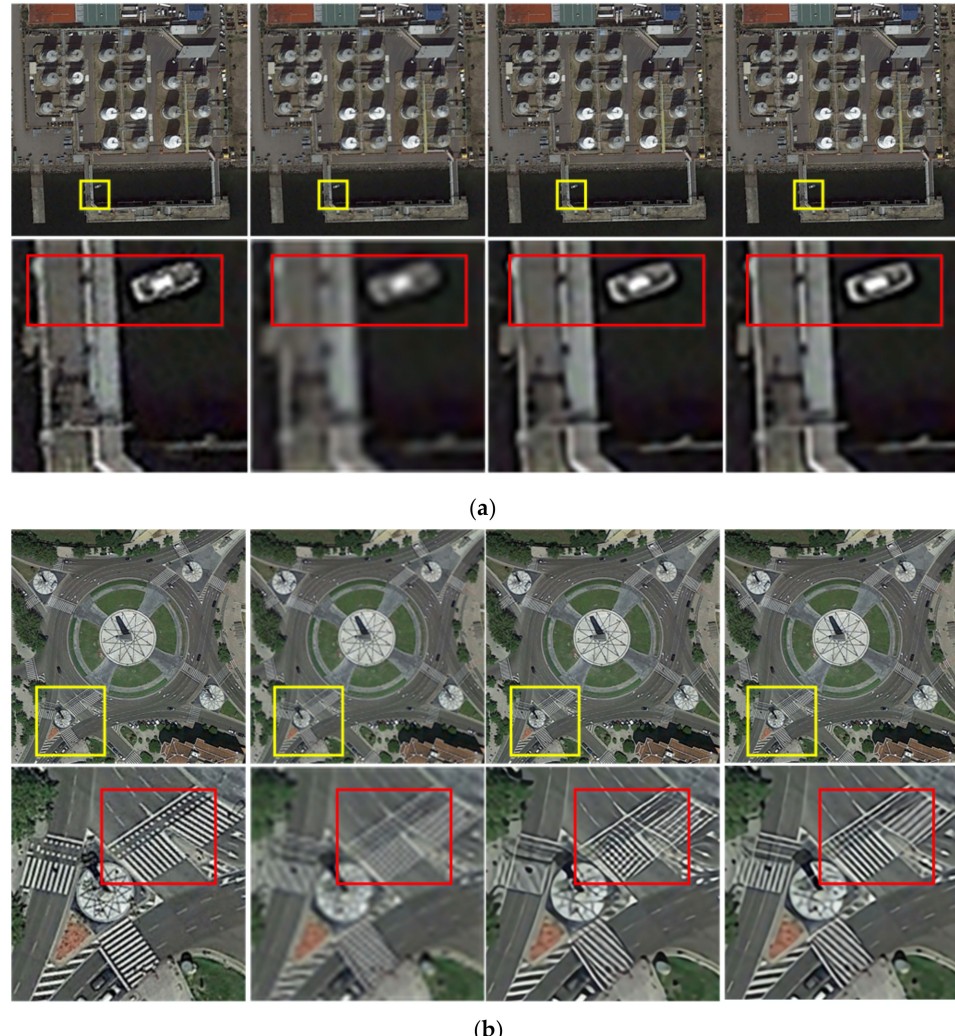

(**a**)

(**b**)

**Figure 11.** *Cont.*

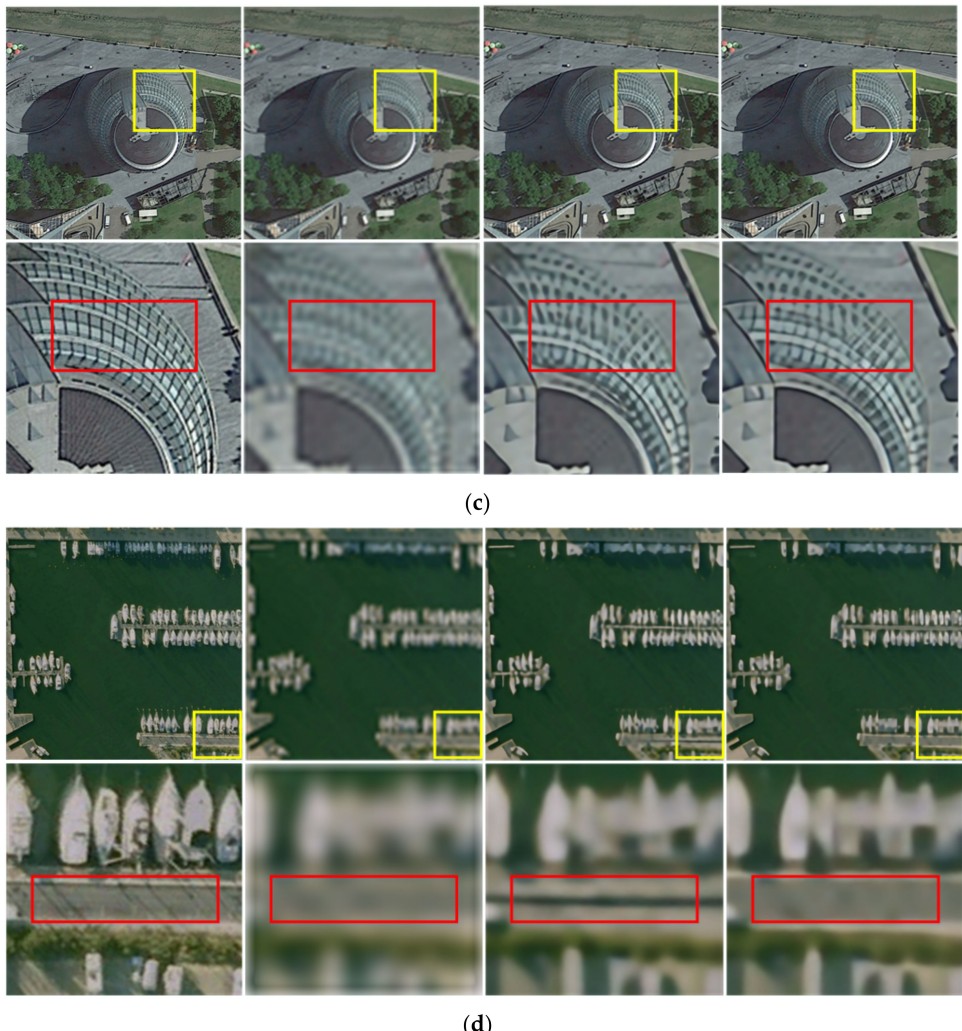

**Figure 11.** Comparison of the reconstruction effect of the PMSRN with different training sets (DIV2K and AID) in each group of images for all *r* values. (Left to right) HR, Bicubic, PMSRN (DIV2K), and PMSRN (AID) images. Compared with the other methods, the SR image obtained using the PMSRN (AID) is closer to the HR. The up-sampling factors *r* value is (**a**) 2, (**b**) 3, (**c**) 4, and (**d**) 8.

For 2× (*r* = 2), the quality of the reconstructed images obtained using the PMSRN (DIV2K) and PMSRN (AID) is much higher than that of the LR images. Accordingly, the PMSRN (AID) clearly shows more detailed information about the bow deck in the red frame. For 3× (*r* = 3), 4× (*r* = 4), and 8× (*r* = 8), the reconstruction effects of the PMSRN (DIV2K) and PMSRN (AID) in the red box are significantly different. It can be clearly observed that the SR images acquired using the PMSRN (AID) are closer to the HR images. Therefore, the PMSRN (AID) was used as the final SR reconstruction network.

## 4. Discussion

All the above experimental results are based on the down-sampling of HR images to obtain LR images. However, in practical applications, LR remote-sensing images are directly collected by sensors. Therefore, it is necessary to discuss further the SR reconstruction effect of the proposed method on actual LR images.

For this reason, in the absence of corresponding high-spatial-resolution images, we firstly selected a remote-sensing image with a spatial resolution of 16 m and randomly selected four areas for experimentation. The methods based on the Bicubic model, MSRN, and PMSRN were used to perform SR reconstruction. Figure 12 presents the results of SR reconstruction with different *r* values.

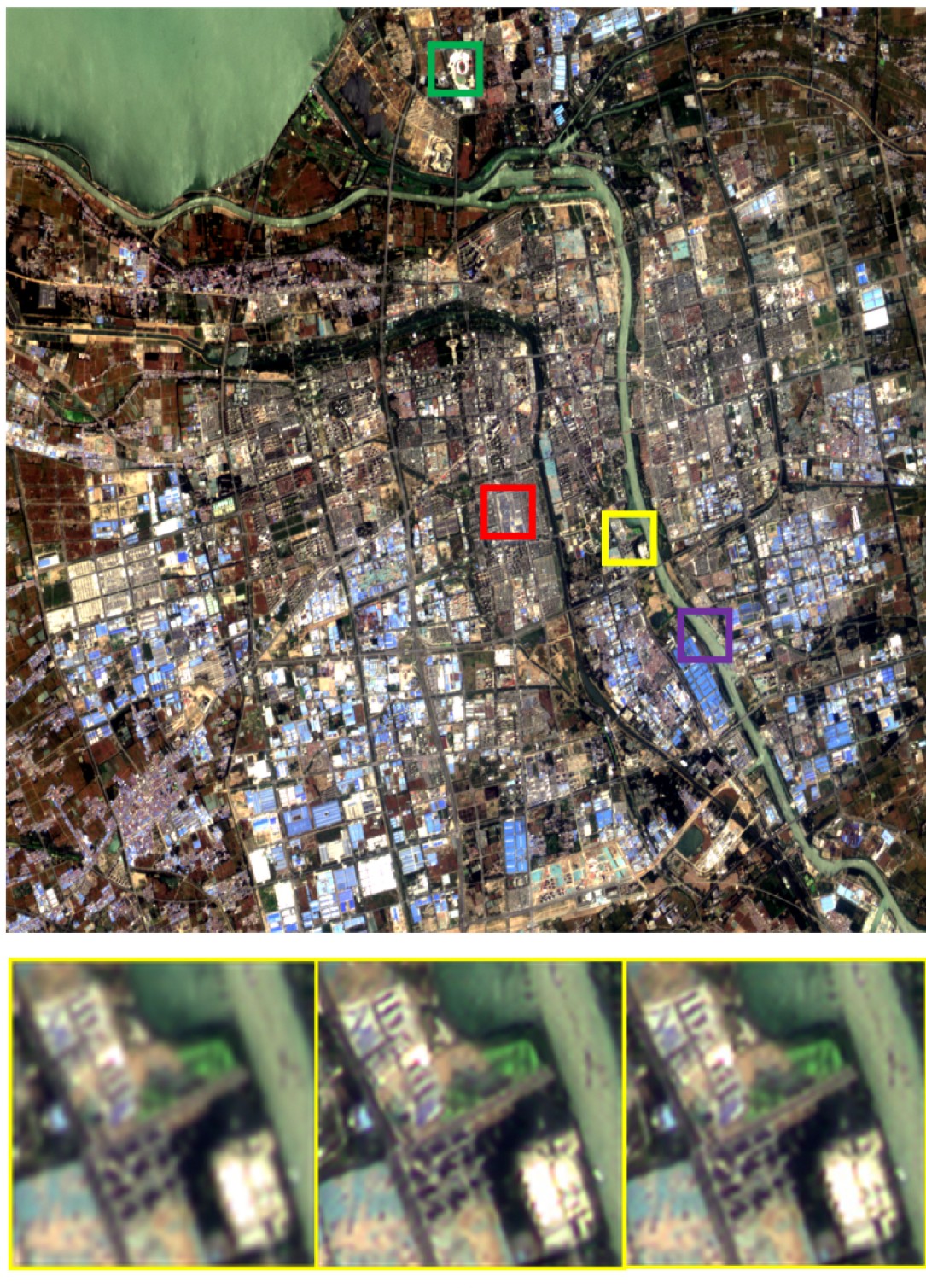

(**a**)

**Figure 12.** *Cont.*

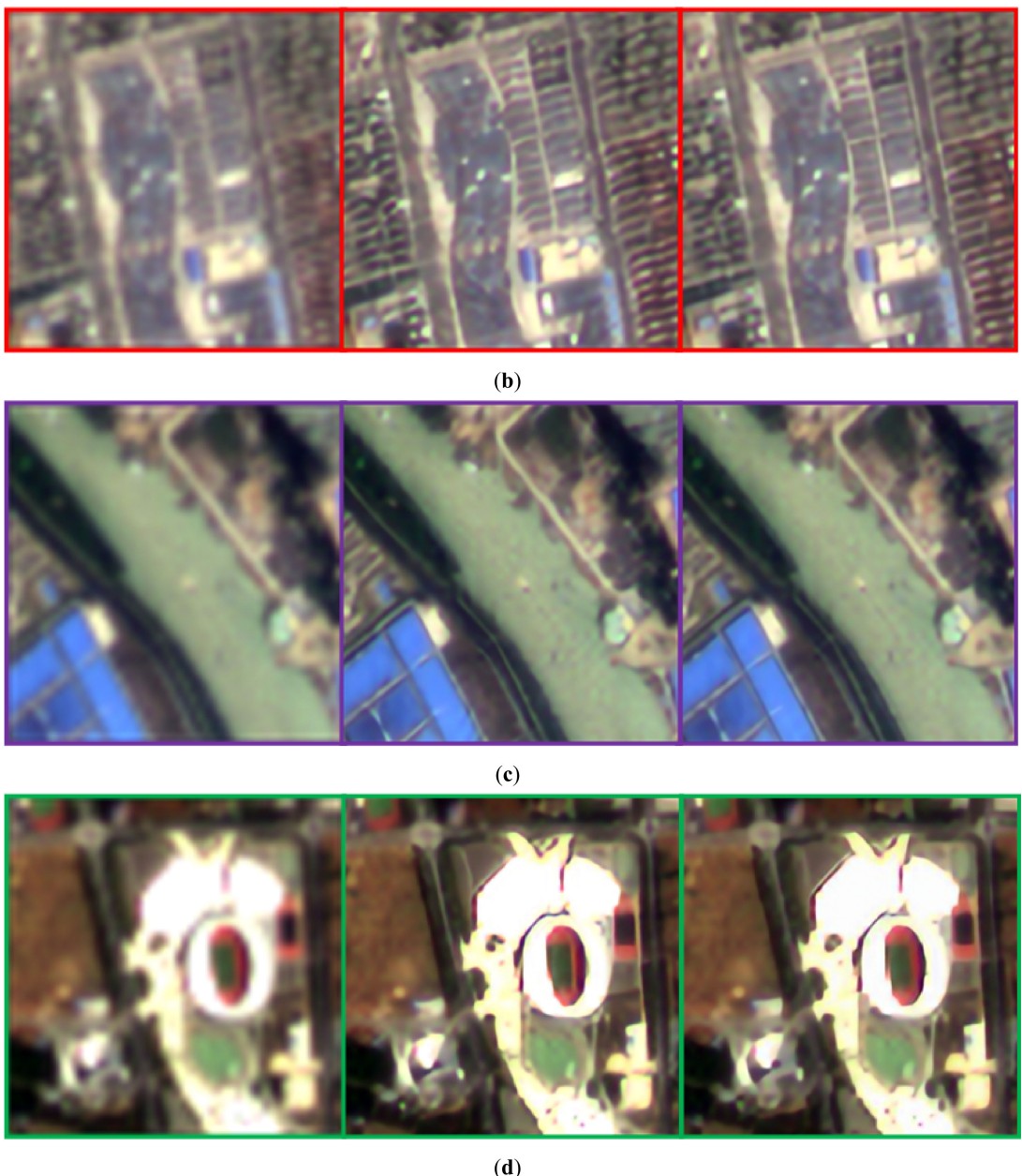

**Figure 12.** Performance comparison of the Bicubic model, MSRN, and PMSRN when applied to remote-sensing images in each group of pictures, for all *r* values. From left to right are the Bicubic, MSRN, and PMSRN results. Compared with other methods, the PMSRN SR image exhibits the best edge detail. The up-sampling factors *r* values are (**a**) 2, (**b**) 3, (**c**) 4, and (**d**) 8.

Owing to the lack of corresponding high-spatial-resolution images, the reconstruction effects can only be evaluated by comparing the SR image definition of the three methods. Based on observation, it was found that the SR image quality based on deep learning (MSRN and PMSRN) is better than that of the Bicubic method. Compared with the MSRN SR image, the PMSRN SR image exhibits more details, fuller colour, and clearer contour stripes.

In addition, ignoring the fact that different wavelengths have different reflectivities in the same area [31], we selected different wavebands belonging to the same remote-sensing image to combine them and obtain HR images with a spatial resolution of 1 m and LR images with spatial resolutions of 2 m and 4 m. We randomly selected two areas for experimentation, as shown in Figure 13.

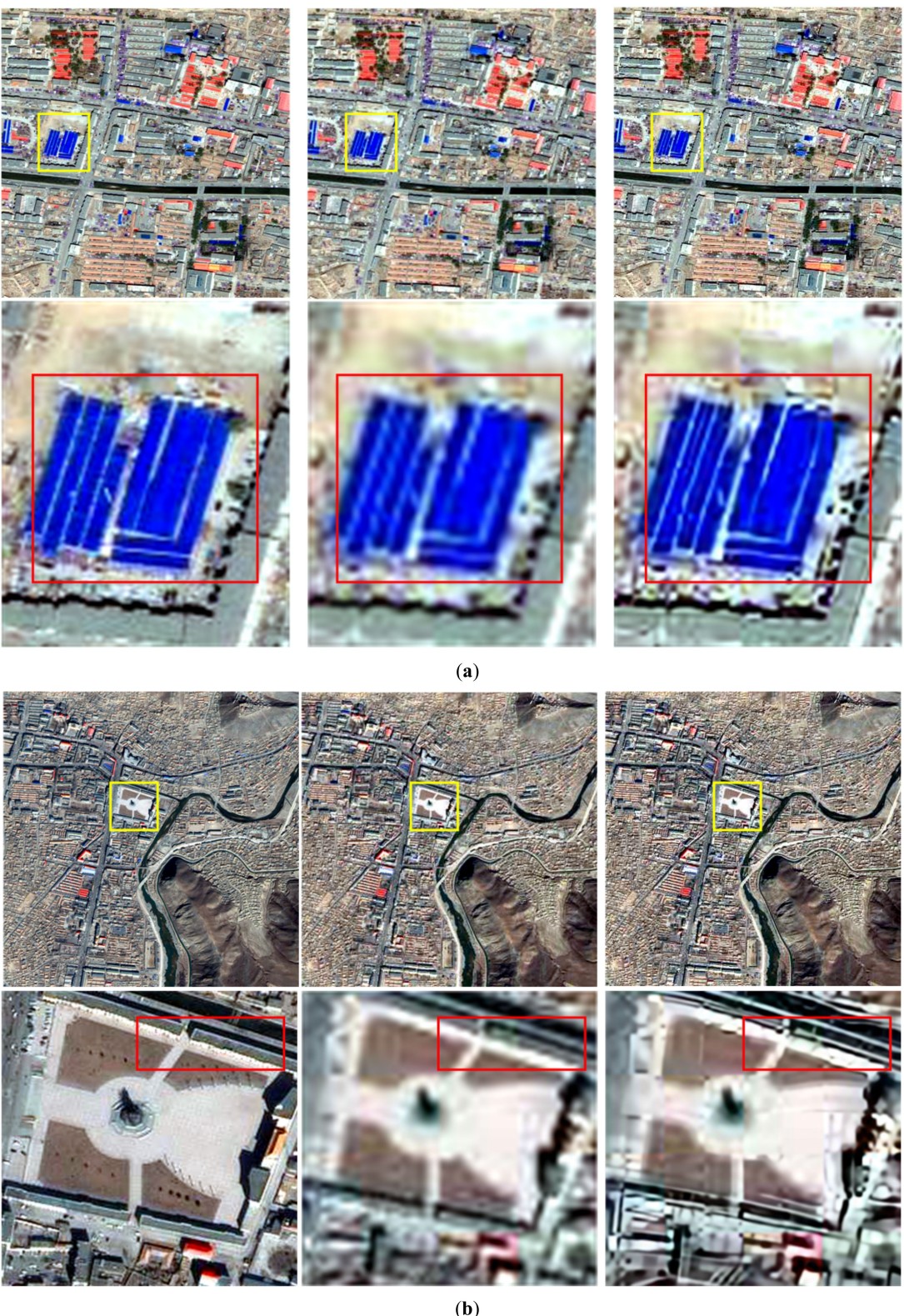

**Figure 13.** Selection of different bands belonging to the same remote-sensing image to combine them, yielding a spatial resolution of 1 m for the high-resolution (HR) image and spatial resolutions of 2 m and 4 m for the low-resolution (LR) images. Comparison of the SR reconstruction effects of the Bicubic model and PMSRN. (Left to right) Images with 1 m spatial resolution, Bicubic model results, and PMSRN outcomes. The up-sampling factors *r* value is (**a**) 2 and (**b**) 4.

The images on the left of Figure 13a,b show images with a spatial resolution of 1 m, which are only used as HR images for visual comparison. The images in the middle of

Figure 13a,b present SR images reconstructed using the Bicubic model, and the images on the right of Figure 13a,b depict SR images reconstructed by the PMSRN. It can be observed that the outline of SR images reconstructed by the PMSRN is clearer than that of Bicubic and closer to the images on the left of Figure 13. Because the ground features are too fuzzy and complex, the reconstructed image in the images on the right of Figure 13b is not as good as that in Figure 13a. However, the definition of the image on the right of Figure 13b is significantly higher than that of the image in the middle of Figure 13b. These results further verify that the PMSRN is of significant value in remote-sensing image research.

## 5. Conclusions

In this report, we proposed an efficient SR reconstruction network called the PMSRN, which is an improved method of the MSRN. The main modules of the PMSRN include MSDRBs and a CB. MSDRBs use numerous residual networks both internally and externally, which can enhance the detection ability of image features on multiple scales and fully utilise image feature information. Furthermore, the network introduces global and local CB when reconstructing SR images, which helps improve the network stability, prevent information loss, and improve the use of original LR image feature information. In addition, comparison of the reconstruction effects of the PMSRN when trained using DIV2K and AID data conclusively indicated that the AID training set is more suitable for remote-sensing image reconstruction. Comparison of the remote-sensing SR images reconstructed by PMSRN (AID) with the high-spatial-resolution remote-sensing images acquired by a satellite indicated that PMSRN (AID) yielded satisfactory results.

Compared with other networks, the PMSRN achieved impressive reconstruction performance on RGB data sets that ignore noise. In future work, it will be necessary to investigate the effects of noise on SR reconstruction as well as the SR reconstruction of non-RGB band images.

**Author Contributions:** Conceptualization, H.H.; methodology, H.H. and P.L.; software, P.L.; validation, P.L., N.Z.; formal analysis, P.L.; investigation, H.H., P.L. and N.Z.; resources, H.H., C.W., Y.X. (Yong Xie); writing—original draft preparation, P.L.; writing—review and editing, H.H., P.L., C.W. and Y.X. (Yaqin Xie); visualization, H.H. and P.L.; supervision, H.H., C.W., D.X. and Y.X. (Yaqin Xie); project administration, H.H. All authors have read and agreed to the published version of the manuscript.

**Funding:** This work was supported in part by the National Natural Science Foundation of China (NSFC) under Grants 41671345 and 62001238.

**Data Availability Statement:** The data presented in this study are available on request from the corresponding author.

**Conflicts of Interest:** All authors have reviewed the manuscript and approved submission to this journal. The authors declare that there is no conflict of interest regarding the publication of this article and no self–citations included in the manuscript.

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
