# Peer review of "End-to-End Super-Resolution for Remote-Sensing Images Using an Improved Multi-Scale Residual Network"

_remotesensing, doi:10.3390/rs13040666_

Round 1

Reviewer 1 Report

Dear authors

Please find the attached comment.

Stay safe,

Reviewer 2 Report

Review of “Pyramidal Multi-scale Residual Network for Super-resolution Remote-sensing Imaging” by Hai Huan, Pengcheng Li , Nan Zou , Chao Wang, Yaqin Xie, Yong Xie and Dongdong Xu.

Generally, the paper is well organized and written.  I generally understand the methodology, but the text is full of jargon (especially in the sections explaining the methodology) making it difficult to follow for someone not intimately involved in this type of research.  Figures and captions need some work to help the reader follow.  Make it easier for the reader to follow your logic progression.  If you make it too hard to follow, you will lose the reader.

  • Abstract: lines 25-26:  “the reconstructed super-resolution image has improved in terms of the evaluation index and visual effect.”  What evaluation index?  Is a commonly used index?  If so, give numerical improvement.  “Visual effect” is kind of a weak explanation of improvement.  Can you quantify the improvement over the most commonly used method?
  • This method is primarily used to sharpen “blurry” images and works best when the same area is imaged at different resolutions. Since surfaces in remotely sensed image data almost always change over time (compositional change, surface moisture, vegetation cover, illumination changes (diurnal and seasonal), It seems to me, that these different resolution images would have to be collect at the same time.  Is that correct?
  • What are the limitations for application of this methodology for remote sensing data? What image data is needed?  Does image data for a particular scene need to be acquired at the same time?  Will it work for images taken at different look angles of the same surface?  Will it work for multispectral or hyperspectral image data? (see next comment)
  • You need a scene imaged at different resolution for this methodology. Do they all need to be imaged at the same wavelength?  My guess is that they do.  For example, ASTER collects image 14 bands of image data at three resolutions (3 visible bands at 10 meter resolution, 6 near-infrared bands at 30 meter resolution, and 5 thermal bands at 90 meter resolutions).  Although the 14 band images are of the same area, taken at the same time, they are at different wavelengths.  So, at a particular wavelength, an area in one image may be bright  but darker in another image at a different wavelength. Could this methodology be used to “sharpen” the 30 and 90 meter image data and retain the band-specific reflectance data?  I suspect than this situation is more complex than you address here.  If not, using image data like ASTER to sharpen the lower resolution bands would greatly improve your results.
  • Materials and Methods: You describe MSRN, MSRB, and PMSRN (your model).  This is pretty confusing.  In your approach. It appears that you use the MSRN (or is PMSRN a modification of this?) and a modified MSRB (MSDRB).  I think you need a paragraph at the beginning of this section to clearly and simply explain to the reader how all these models fit together.
  • How about a paragraph that explicitly states the minimum image requirement needed to successfully apply this methodology. This could be in the conclusions when you could state that with only the following image data, you can increase the spatial clarity by x%.
  • Conclusions:
    1. Line 500: “In this study, an efficient SR reconstruction network (PMSRN) was proposed.”  Propose change:  “In this paper, we propose a SR reconstruction network (PMSRN) that is both efficient and effective in …….  This approach is an improvement over __________ .”  State when this approach should be used as well as when not to use it. 
    2. Line 514: “generalisation”  should be generalization
    3. Line513-515: “However, the network trained by the dataset constructed in this manner had poor generalisation ability for LR real-world images with fuzzy kernels and noise.” What are you trying to say here?  Use something else?  Beware of ____?  Works best under certain conditions?

Figure Captions:

  • Figures 1 thru 4 - These captions are marginally useful, if at all. How about
  • Figures 7-9: Add text to tell the reader what you want them to see.  Help the reader follow your reasoning and your results. Blue and green lines mimic each other or what?
  • Figure 10 - 13. Would be helpful if you labeled the rows instead of using (a), (b)…  Make it easier on the reader to follow.  Also, add a sentence to the captions stating what you want them to get from the figure.  For example, ”Note that the SR (model sharpened image) improves the image clarity over the low-resolution (LR) image.”

Reviewer 3 Report

In this paper, a pyramidal multi-scale residual network for super resolution reconstruction is proposed. Also, a multi-scale dilation residual block is proposed, inspired in a previous work.

The article is difficult to read, so I think it should be reviewed and improved before publication, taking into account the following observations:

1. The abstract states that: "PMSRN introduces a new type of Res2Net residual block". However, this block was proposed in

Gao, S., Cheng, M. M., Zhao, K., Zhang, X. Y., Yang, M. H., & Torr, P. H. (2019). Res2net: A new multi-scale backbone architecture. IEEE transactions on pattern analysis and machine intelligence.

Please correct.

2. Review or explain the sentence: "In addition, a complementary block is added to improve the problem by ignoring useful original information after filtering several convolution layers"

3. Line 31, review or justify why the image resolution is a "performance parameter".

4. Lines 43-46: The sentence: "An end-to-end convolutional neural network (CNN) model was constructed, and the mapping relationship between LR and HR was learnt using the open dataset training network to identify the optimal solution".

I think that this sentence is out of context, since this apart is talking about state of the art (i.e. it is not talking about the proposed work).

5. Line 68: "the effect of reproduction is far from the results of this study". This section is talking about the drawbacks in the state of the art, and not of the proposed work. Please, review it.

6. The contribution of the study states that "PMSRNs are easier to train", however, the experiment section does not include anything related to this statement. Please review.

7. Line 122: "Li et al. in 2018". Please unify the cites of the references.

8. Line 142 and Figure 2, include HFFS. However, this module is not explained in the paper. Please include it in the paper.

9. Section "Multi-dilation Rate Dilated Convolution Group". It is necessary to articulate the explanation of this sub-section with the explanation of the method.

10. Briefly explain what the PixelShuffle consists of.

11. Please review the English language and style througout the manuscript.

12. Lines 263-264: "and each pixel size is 600 × 600". I think that this refers to image size (i.e. "and the image size is 600 x 600"). Please review and correct similar errors throughout the manuscript.

13. Lines 273-274: "Before training, each training block". Maybe, it refers to training image? (i.e. "before training, each training image")

14. Line 287. "HFF module": this module is not explained in the paper. Please include it in the paper.

15. Section 3.1.2 talks about pre-training model. However, throughout the paper, nothing is said about pre-trained models.

16. Review the sentence in lines 331-333: "Although in the training process, the PMSRN and MSRN network had outliers because the final network model parameters w and b assumed the weight value at the highest point of the PSNR curve." This sentence is incomplete.

17. Lines 352-393: This paragraphs summarizes the data presented in Table 1. It would be necessary to present a more in-depth analysis (beyond summarizing the data in the table).

18. In relation to remote sensing, different aspects must be analyzed. For example, what happens with bands other than RGB, or what happens when you have bands in the image with different spatial resolution than RGB bands?

Round 2

Reviewer 1 Report

1- Please edit the English language. specifically your introduction. Introduction  has not written well. there is no consistency between items and there are several typos. a perfect introduction is necessary for such article.

2- in Section. 2, please make a connection between subsections. specifically network architecture and scale fusion. equations must follow same symbols and parameters. 

regards, 

Reviewer 3 Report

The suggested changes have been made. However, there are still some things to enhance.

  1. The sentences in Lines 33-40 are confusing. For example, "Please note that resolution is different from spatial resolution". Remember that you can have spatial resolution, spectral resolution, radiometric resolution, or temporal resolution. Please review.

Some explanations about this:

https://www.nrcan.gc.ca/maps-tools-publications/satellite-imagery-air-photos/remote-sensing-tutorials/satellites-sensors/spatial-resolution-pixel-size-and-scale/9407 

https://www.nrcan.gc.ca/maps-tools-publications/satellite-imagery-air-photos/remote-sensing-tutorials/satellites-sensors/spectral-resolution/9393 

https://www.nrcan.gc.ca/maps-tools-publications/satellite-imagery-air-photos/remote-sensing-tutorials/satellites-sensors/radiometric-resolution/9379 

https://www.nrcan.gc.ca/maps-tools-publications/satellite-imagery-air-photos/remote-sensing-tutorials/satellites-sensors/temporal-resolution/9365

  1. The definition given in Lines 52-54, should be enhanced in order to clarify if it refers to Deep learning definition, or if it refers to the application of Deep learning to super resolution.

  1. Please replace the word "whose", in the sentence "The proposed PMSRN is easier to train than other networks, whose number of parameters is only 43.33% of that of EDSR, and the module is independent and easy to migrate to other networks for learning".

i.e.: "The proposed PMSRN is easier to train than other networks, since its number of parameters is only 43.33% of that of EDSR, and the module is independent and easy to migrate to other networks for learning"

  1. Please review the English language and style throughout the manuscript.
